# Algal-fungal symbiosis leads to photosynthetic mycelium

Zhi-Yan Du[1,2,3], Krzysztof Zienkiewicz[2,4,5], Natalie Vande Pol[6], Nathaniel E Ostrom[7,8], Christoph Benning[1,8,2,3]*, Gregory M Bonito[6,8,9]*

[1]Department of Energy-Plant Research Laboratory, Michigan State University, East Lansing, United States; [2]Department of Biochemistry and Molecular Biology, Michigan State University, East Lansing, United States; [3]Department of Plant Biology, Michigan State University, East Lansing, United States; [4]Department of Plant Biochemistry, Albrecht-von-Haller-Institute for Plant Sciences, Georg-August-University, Göttingen, Germany; [5]Centre of Modern Interdisciplinary Technologies, Nicolaus Copernicus University in Toruń, Toruń, Poland; [6]Department of Microbiology and Molecular Genetics, Michigan State University, East Lansing, United States; [7]Department of Integrative Biology, Michigan State University, East Lansing, United States; [8]DOE Great Lakes Bioenergy Research Center, Michigan State University, East Lansing, United States; [9]Department of Plant, Soil and Microbial Sciences, Michigan State University, East Lansing, United States

**Abstract** Mutualistic interactions between free-living algae and fungi are widespread in nature and are hypothesized to have facilitated the evolution of land plants and lichens. In all known algal-fungal mutualisms, including lichens, algal cells remain external to fungal cells. Here, we report on an algal–fungal interaction in which *Nannochloropsis oceanica* algal cells become internalized within the hyphae of the fungus *Mortierella elongata*. This apparent symbiosis begins with close physical contact and nutrient exchange, including carbon and nitrogen transfer between fungal and algal cells as demonstrated by isotope tracer experiments. This mutualism appears to be stable, as both partners remain physiologically active over months of co-cultivation, leading to the eventual internalization of photosynthetic algal cells, which persist to function, grow and divide within fungal hyphae. *Nannochloropsis* and *Mortierella* are biotechnologically important species for lipids and biofuel production, with available genomes and molecular tool kits. Based on the current observations, they provide unique opportunities for studying fungal-algal mutualisms including mechanisms leading to endosymbiosis.

DOI: https://doi.org/10.7554/eLife.47815.001

*For correspondence:
benning@msu.edu (CB);
bonito@msu.edu (GMB)

Competing interests: The authors declare that no competing interests exist.

## Introduction

Mutualistic symbioses are defined as those in which partners interact physically and metabolically in mutually beneficial ways. Mutualisms underlie many evolutionary and ecological innovations including the acquisition of plastids and mitochondria, and evolution of symbiotic mutualisms such as mycorrhizas, lichens and corals (*Little et al., 2004*; *Service, 2011*; *Tisserant et al., 2013*; *Spribille et al., 2016*). An understanding of the underlying principles that govern microbial mutualisms informs microbial ecology and efforts to engineer synthetic microbiomes for biotechnological applications (*Egede et al., 2016*).

Terrestrialization of Earth has been associated with lineages of early diverging fungi belonging to the Mucoromycota. However, recent analyses indicate that fungal colonization of land was associated with multiple origins of green algae prior to the origin of embryophytes (*Lutzoni et al., 2018*).

**eLife digest** Yeast, molds and other fungi are found in most environments across the world. Many of the fungi that live on land today form relationships called symbioses with other microbes. Some of these relationships, like those formed with green algae, are beneficial and involve the exchange carbon, nitrogen and other important nutrients. Algae first evolved in the sea and it has been suggested that symbioses with fungi may have helped some algae to leave the water and to colonize the land more than 500 million years ago.

A fungus called *Mortierella elongata* grows as a network of filaments in soils and produces large quantities of oils that have various industrial uses. While the details of *Mortierella*'s life in the wild are still not certain, the fungus is thought to survive by gaining nutrients from decaying matter and it is not known to form any symbioses with algae.

In 2018, however, a team of researchers reported that, when *M. elongata* was grown in the laboratory with a marine alga known as *Nannochloropsis oceanica*, the two organisms appeared to form a symbiosis. Both the alga and fungus produce oil, and when grown together the two organisms produced more oil than when the fungus or algal cells were grown alone. However, it was not clear whether the fungus and alga actually benefit from the symbiosis, for example by exchanging nutrients and helping each other to resist stress.

Du et al. – including many of the researchers involved in the earlier work – have now used biochemical techniques to study this relationship in more detail. The experiments found that there was a net flow of carbon from algal cells to the fungus, and a net flow of nitrogen in the opposite direction. When nutrients were scarce, algae and fungi grown in the same containers grew better than algae and fungi grown separately. Further, *Mortierella* only obtained carbon from living algae that attached to the fungal filaments and not from dead algae. Unexpectedly, further experiments found that when grown together over a period of several weeks or more some of the algal cells entered and lived within the filaments of the fungus. Previously, no algae had ever been seen to inhabit the living filaments of a fungus.

These findings may help researchers to develop improved methods to produce oil from *M. elongata* and *N. oceanica*. Furthermore, this partnership provides a convenient new system to study how one organism can live within another and to understand how symbioses between algae and fungi may have first evolved.

DOI: https://doi.org/10.7554/eLife.47815.002

Research indicating that plants were genetically pre-adapted for symbiosis with fungi, has renewed interest in fungal-algal associations (*Delaux et al., 2015*; *Spatafora et al., 2016*).

The most well-known mutualisms that exist between algae and fungi are lichens, which were estimated to radiate 480 million years ago (*Lutzoni et al., 2018*). Lichen symbiosis is adaptive in that it allows mycobiont and photobiont symbionts to survive in habitats and environments that would otherwise be uninhabitable by either species growing alone, such as on a rock outcrop or in a desert crust. Lichenized fungi have been shown to have multiple independent origins in Ascomycota and Basidiomycota, and are themselves meta-organisms that include communities of chlorophyte algae, cyanobacteria, in addition to basidiomyceteous yeasts (*Spribille et al., 2016*).

Nutrient exchange often underlies mutualisms between photobionts and mycobionts. For example, reciprocal transfer of carbon and nitrogen was shown for synthetic consortia composed of *Chlamydomonas reinhardtii* and a diverse panel of ascomycete fungi, demonstrating a latent capacity of ascomycetous yeasts and filamentous fungi to interact with algae (*Hom and Murray, 2014*). In a separate study, the filamentous ascomycetous fungus *Alternaria infectoria* was demonstrated to provision nitrogen to *C. reinhardtii* in a long-lived bipartite system, whereby the nitrogen-starved alga responded favorably to the growing fungus (*Simon et al., 2017*). A non-lichen algal-fungal mutualism was described involving the chytrid fungus *Rhizidium phycophilum* and the green alga *Bracteacoccus* providing evidence that early diverging fungi have evolved mutualisms with algae based on solute exchange (*Picard et al., 2013*). However, in all known examples of fungal-algal symbioses algal cells remain external to fungal hyphae and are not known to enter living fungal cells.

While studying a synthetic co-culture composed of two biotechnologically important oil-producing organisms, the soil fungus *Mortierella elongata* and the marine alga *Nannochloropsis oceanica*, we observed an interaction between fungal and algal cells that led to changes in metabolism of both partners (*Du et al., 2018a*). This biotrophic interaction showed high specificity and resulted in close physical contact of partners, with the eventual incorporation of functional algal cells within fungal mycelium. Here, we describe this apparent symbiosis in detail. We further demonstrate through isotope tracer experiments that bidirectional nutrient exchange underlies the described algal-fungal interactions.

## Results

### Interaction between *N. Oceanica* and *M. elongata* after short-term co-culture

*Nannochloropsis oceanica* cells flocculated in dense clusters around *M. elongata* mycelium when they were incubated together (*Figure 1A and B*). After 6 days co-cultivation, scanning electron microscopy (SEM) revealed a wall-to-wall fungal-algal interface between the organisms grown in co-culture, (*Figure 1C*) with the morphology of *N. oceanica* cells differing from those of cells grown in the absence of fungus. Specifically, SEM images showed that *N. oceanica* cells incubated alone in f/2 medium have a smooth outer layer (*Figure 1D* and *Figure 1—figure supplement 1A*), which was fragmented or lacking after co-culture with *M. elongata* AG77 and fibrous extensions underneath the smooth outer layer were exposed (*Figure 1E* and *Figure 1—figure supplement 1B*). While it is possible that the fibrous extensions have been elicited by the contact with the fungus, remnant pieces of the outer coat covering the underlying extensions are evident in our observations (*Figure 1E and F* and *Figure 1—figure supplement 1B*). Therefore, it seems likely that these extensions are present underneath the outer smooth layer. In addition, the fibrous extensions appeared to contribute to anchoring the algae to hyphae and irregular tube-like extensions were formed between the two interacting cell types (*Figure 1F*). Further SEM revealed that the fibrous extensions only were exposed in *N. oceanica* cells that were in physical contact with live fungal hyphae. *N. oceanica* cells maintained a smooth outer wall covering (*Figure 1—figure supplement 1*) when they were co-cultured without physical contact with live fungi, and when they were co-cultured with *M. elongata* mycelium that had been killed in a 65°C water bath. We demonstrated that a combination of enzymes, including 4% fungal hemicellulase and 2% driselase, could partially digest the outer smooth cell wall layer of *N. oceanica* and expose the fibrous extensions to mimic the morphological change observed during the physical interaction between live *M. elongata* and *N. oceanica* cells (*Figure 1—figure supplement 2*).

### Carbon and nitrogen transfer between *N. oceanica* and *M. elongata*

To test whether nutrient, that is carbon or nitrogen exchange underlies the interaction between *M. elongata* and *N. oceanica*, we conducted a series of tracer experiments using reciprocally $^{14}$C- and $^{15}$N-labeled algal and fungal partners. For carbon exchange assays, algal cells were labeled with [$^{14}$C]-sodium bicarbonate and were co-cultivated with actively growing non-labeled fungal hyphae for 1 week in flasks. Conversely, fungal hyphae were grown in either [$^{14}$C]-glucose- or [$^{14}$C]-acetate-containing medium for labelling. Labeled fungi were then co-incubated with non-labeled algal cells in flasks that allowed the two organisms to interact physically. Co-cultured algal and fungal cells were separated from each other by cellulase digestion and mesh filtration (*Figure 2—figure supplement 1A–E*). Algal and fungal cells were collected and analyzed for $^{14}$C exchange, separately. Isotope analyses indicated that a significant amount of $^{14}$C-carbon was transferred from the alga to the fungus, and nearly 70% of the total transferred $^{14}$C-carbon was incorporated into the fungal lipid pool, with the remaining incorporated into free amino acids (FAAs), proteins, soluble compounds, and carbohydrates (*Figure 2A*, left). Similarly, $^{14}$C-carbon transfer was observed from the labeled fungus to its algal recipient (*Figure 2A*, right). Fractions of algal cells attached to the fungal hyphae acquired more $^{14}$C than unattached cells sampled in the supernatant (*Figure 2A* and *Figure 2—figure supplement 1F and G*).

To assess whether a physical interaction is required for carbon exchange between the photosynthetic alga and the putative fungal heterotroph, we used membrane inserts to physically separate

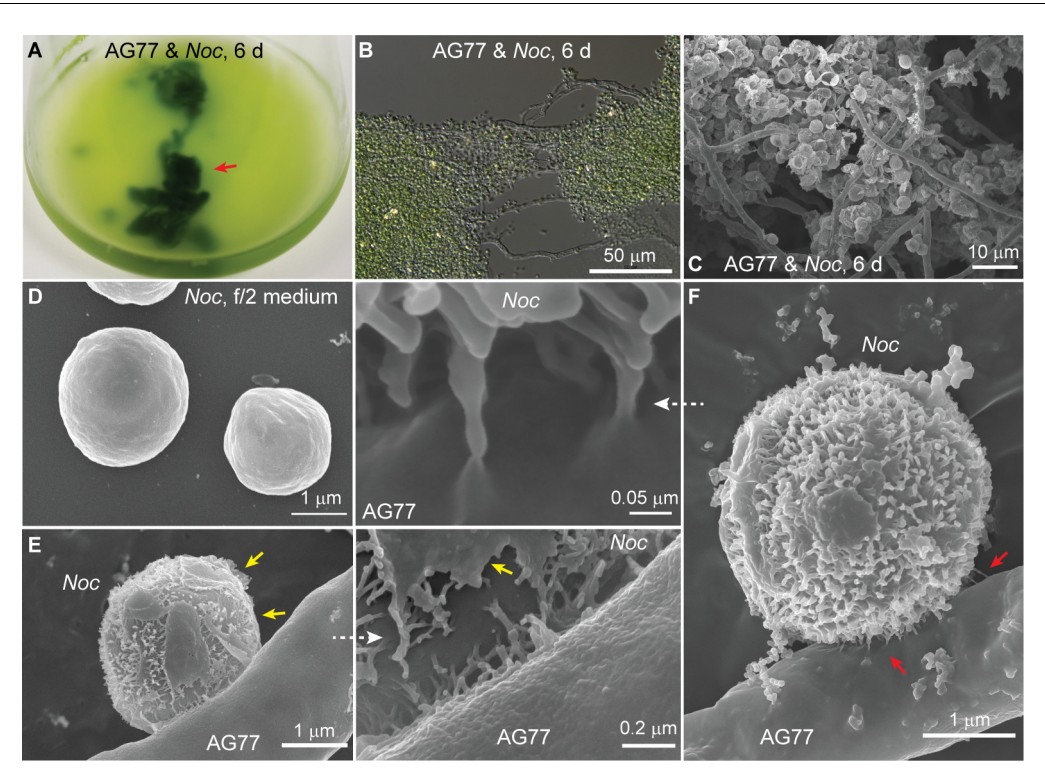

**Figure 1.** Interaction between *N.oceanica* and *M. elongata* cells. (**A**) Co-cultivation of *M. elongata* AG77 and *N. oceanica* (*Noc*) in flasks for 6 days. Green tissues indicated by the red arrow head are aggregates formed by AG77 mycelium and attached *Noc* cells. (**B**) Differential interference contrast micrographs of the green tissues shown in (**A**). A large number of *Noc* cells are trapped by AG77 mycelium. (**C–F**) Scanning electron microscopy images of alga-fungus interaction. (**C**) N*oc* cells stick to the fungal mycelium after 6-day co-culture. (**D**) *Noc* controls grown in f/2 medium alone have smooth surface. (**E**) A *Noc* cell adheres to an AG77 hypha by the outer surface with fibrous extensions, which were exposed after break of the original out layer. Yellow arrows indicate the residues of the out layer. (**F**) A *Noc* cell anchored to the AG77 hypha by the fibrous extensions. Red arrows indicate irregular tube-like extensions of the *Noc* cell wall connected to the surface of fungal cell wall.

DOI: https://doi.org/10.7554/eLife.47815.003

The following figure supplements are available for figure 1:

**Figure supplement 1.** Interaction between *N.oceanica* cells and *M. elongata* AG77 hyphae.

DOI: https://doi.org/10.7554/eLife.47815.004

**Figure supplement 2.** *N.oceanica* cells were treated with different enzymes to mimic the expose of fibrous extensions after co-culture with fungi.

DOI: https://doi.org/10.7554/eLife.47815.005

reciprocally $^{14}$C-labeled algal and fungal partners (*Figure 2—figure supplement 2A–C*). We observed that the physical contact between the algae and fungus is essential for $^{14}$C-carbon transfer to the fungus (*Figure 2B and C*) but is not necessary for $^{14}$C-carbon transfer to the algal cells (*Figure 2B and D* and *Figure 2—figure supplement 2D*).

Considering that *Mortierella* is commonly regarded as a saprotroph that acquires carbon from dead organic matter (*Phillips et al., 2014*), we tested whether alga-derived carbon obtained by *M. elongata* was due to the consumption of algal detritus. First, we repeated the $^{14}$C-labeling experiment described above using a 65℃ water bath to kill $^{14}$C-labeled cells prior to algal-fungal reciprocal pairings. We found that *M. elongata* incorporates only a small amount (1.3%) of $^{14}$C-carbon from dead algal cells, compared to $^{14}$C-carbon acquired from living algal cells (12.7%) (*Figure 2C* and *Figure 2—figure supplement 2E*). In contrast, the algal cells attached to fungal hyphae (Att) and those free in the medium (Free) acquired more $^{14}$C-carbon (Att, 2.4%; Free, 15.8%) from dead fungal cells (*Figure 2D*). The total abundance of $^{14}$C-carbon was higher in the free algal cells, because most of

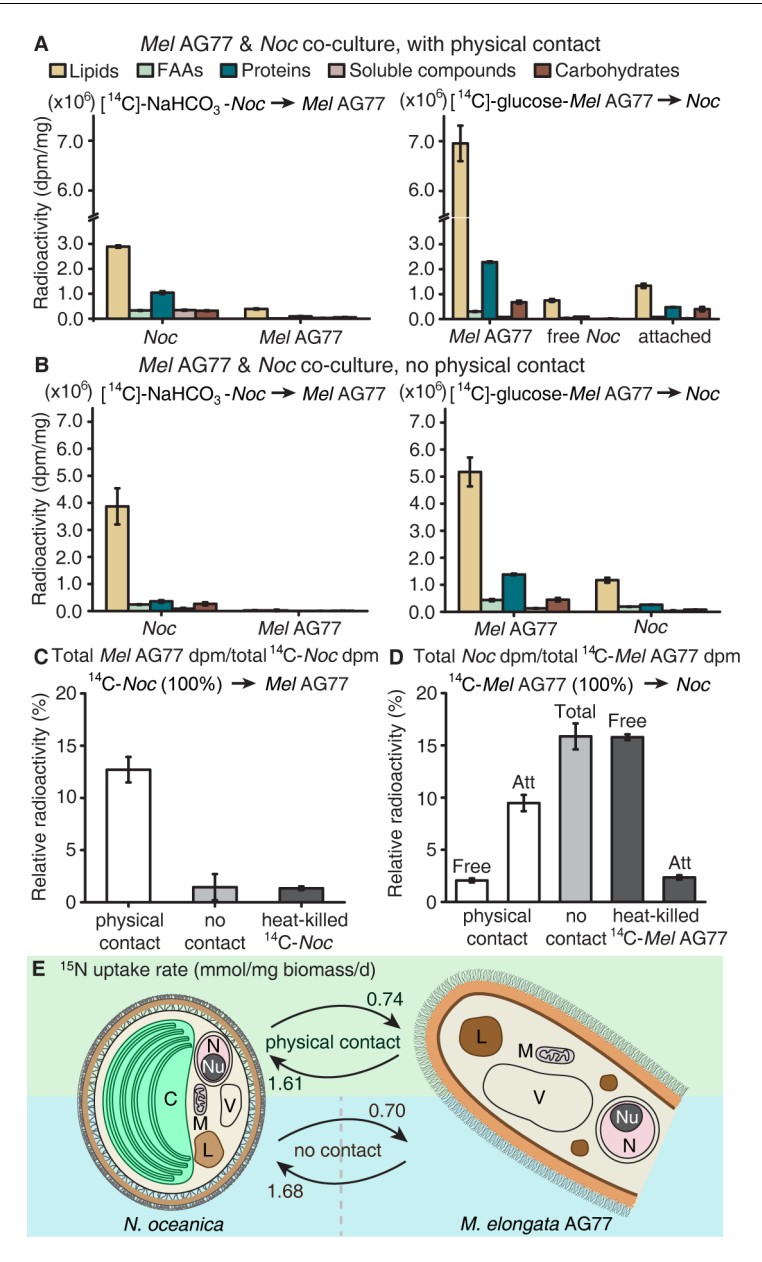

**Figure 2.** Carbon exchange between *N.oceanica* and *M. elongata* AG77. (**A**) Carbon (**C**) transfer from [14C]-sodium bicarbonate (NaHCO₃)-labeled *N. oceanica* (*Noc*) cells to *M. elongata* AG77 (*Mel* AG77, left panel) or from [14C]-glucose-labeled AG77 to *Noc* cells (right panel) after 7-d co-culture in flasks (with physical contact). Radioactivity of 14C-carbon was determined with a scintillation counter (dpm, radioactive disintegrations per minute) and then normalized to the dry weight of samples (dpm/mg biomass). Free *Noc*, unbound *Noc* cells in the supernatant; attached, *Noc* cells separated from AG77-*Noc* aggregates by algal cell wall digestion and mesh filtration; FAAs, free amino acids; soluble compounds, supernatant after acetone precipitation of proteins extracted by SDS buffer. Data are presented as the average of three biological replicates with standard deviation (Means ± SD, n = 3). (**B**) 14C-carbon transfer between *Noc* and AG77 without physical contact. Algae and fungi were incubated in cell-culture plates with filter-bottom inserts (pore size of 0.4 μm) which separate *Noc* cells and AG77 mycelium from each other but allow metabolite exchange during co-culture. Error bars indicate SD of three biological replicates (n = 3). (**C and D**) Relative abundance of 14C-carbon radioactivity in recipient cells compared to 14C-labeled donor cells after 7-d co-culture. (**C**) AG77 relative to [14C]-NaHCO₃-*Noc* (100%). (**D**) *Noc* relative to [14C]-glucose-labeled AG77 (100%). Physical contact, living 14C-labeled cells added to unlabeled cells for co-cultivation in flasks; no contact, samples grown separately in plates with inserts; heat-killed 14C-cells, 14C-labeled *Noc* or AG77 killed by heat treatment at 65°C for 15 min before the addition to unlabeled cells in flasks. Free, unbound *Noc* cells in the

*Figure 2 continued*

supernatant; Att, *Noc* cells attached to AG77 (isolated by algal cell wall digestion and mesh filtration); Total, *Noc* cells grown separately from AG77 in plates and inserts. Error bars indicate SD of three biological replicates (n = 3). (E) Nitrogen (N) exchange between *N. oceanica* (*Noc*) and *M. elongata* AG77 examined by $^{15}$N-labeling experiments. [$^{15}$N]-potassium nitrate-labeled *Noc* cells or [$^{15}$N]-ammonium chloride-labeled AG77 were added to unlabeled AG77 or *Noc* cells, respectively, for 7-day co-culture in flasks (physical contact) or cell-culture plates with inserts (no physical contact). Algae and fungi were separated and weighed (dry biomass) after the co-culture, and their isotopic composition in Atom% $^{15}$N [$^{15}$N/($^{15}$N+$^{14}$N)100%] and N content (%N) were determined using an elemental analyzer interfaced to an Elementar Isoprime mass spectrometer following standard protocols. The N uptake rate of $^{15}$N-*Noc*-derived N ($^{15}$N) by AG77 from and that of $^{15}$N-AG77-derived N by *Noc* cells ($^{15}$N) were calculated based on the Atom% $^{15}$N, %N and biomass. C, chloroplast; N, nucleus; Nu, nucleolus; M, mitochondrion; V, vacuole; L, lipid droplet. Values are the average of three biological repeats.
DOI: https://doi.org/10.7554/eLife.47815.006

The following figure supplements are available for figure 2:

**Figure supplement 1.** Carbon transfer between *N.oceanica* and *M. elongata* AG77 with physical contact.
DOI: https://doi.org/10.7554/eLife.47815.007

**Figure supplement 2.** Carbon and nitrogen exchange between *N.oceanica* and *M. elongata* AG77 without physical contact.
DOI: https://doi.org/10.7554/eLife.47815.008

**Figure supplement 3.** Viability of *N.oceanica* and *M. elongata* AG77 during 7-d co-culture.
DOI: https://doi.org/10.7554/eLife.47815.009

**Figure supplement 4.** Nitrogen exchange between *N.oceanica* and *M. elongata* AG77.
DOI: https://doi.org/10.7554/eLife.47815.010

the *N. oceanica* cells in the medium were free and contained a similar amount of $^{14}$C-carbon per mg compared to attached cells (*Figure 2—figure supplement 2F*). Second, we used confocal microscopy and Sytox Green staining to assess whether fungal and algal cells remained alive during co-culture. Over 95% of algal cells were alive during the period of reciprocal co-cultivation with $^{14}$C-carbon-labeled cells, and no dead fungal cells were observed (*Figure 2—figure supplement 3A–I*). Moreover, the micrographs show that the heat treatment was effective in killing algal and fungal cells (*Figure 2—figure supplement 3C–E*). Together these data indicate that carbon-transfer from the alga to the fungus is dependent upon physical interaction between living partners. In contrast, the algal cells are able to utilize carbon from the fungus grown in the same culture regardless of whether the hyphae are alive, dead or physically connected.

Nitrogen is a major macronutrient that limits net primary productivity in terrestrial and aquatic ecosystems, especially for microalgae such as *N. oceanica* (*Howarth et al., 1997*; *Vieler et al., 2012*; *Zienkiewicz et al., 2016*). To determine whether nitrogen exchange occurs between *M. elongata* and *N. oceanica*, we grew algal cells with [$^{15}$N]-potassium nitrate and the fungus with [$^{15}$N]-ammonium chloride as the sole nitrogen source. The labeled cells were co-cultivated with unlabeled partners for 1 week, and then were separated and analyzed for $^{15}$N. We detected $^{15}$N-nitrogen transfer between algal and fungal partners, irrespective of whether they were in physical contact or not (*Figure 2E* and *Figure 2—figure supplement 4*). Over twice as much $^{15}$N (~1.6 μmol/mg biomass/d) was transferred from the $^{15}$N-fungus to the algal recipient, than from the $^{15}$N-algal cells to the fungus (~0.7 μmol/mg biomass/day; *Figure 2E*), demonstrating a net nitrogen benefit for the alga when co-cultivated with the fungus. The N transfer under conditions of no-contact between the algae and fungi is relatively high compared to the experiment allowing physical-contact, possibly due to the differences in the culturing system. The physical-contact culture was grown in 125-mL flask containing 25 ml medium, while the no-contact culture was incubated in the 6-well culture plates with 5 ml medium in each well, which is a denser culture with the two species only separated by a thin membrane.

## Nutrient deficiency and benefits of co-cultivation for *N. oceanica* and *M. elongata*

To test whether the algae and fungi benefit from the interaction and the exchange of nutrients, we observed growth in macro- and micronutrient-deficient media. During nitrogen or carbon

deprivation in f/2 medium, *N. oceanica* had significantly increased viability when co-cultivated with *M. elongata* (**Figure 3A–C**). No impact of micronutrients was detected. Element analysis of the culture supernatant showed an increase in total organic carbon and dissolved nitrogen when the living *M. elongata* hyphae were incubated alone in f/2 medium (**Figure 3D and E**). This is indicative of extracellular release of nutrients by the hyphae, and may explain why physical contact is not required for the $^{14}$C-carbon transfer from the fungus to the alga. In addition, following 10-day-prolonged incubation in regular f/2 medium *N. oceanica* cells showed significant higher levels of chlorophyll with the presence of *M. elongata* compared to algal cells grown alone, suggesting that the co-cultured algae likely had a higher photosynthetic capacity (**Figure 3—figure supplement 1A**).

On the other hand, since the viability of *M. elongata* was not obviously affected following nutrient deprivation (**Figure 3—figure supplement 1B–F**), the biomass and growth of *Mortierella* were estimated using a fatty acid biomarker that can be readily quantified by gas chromatography (GC), and light microscopy, respectively. It was not practical to directly determine fungal biomass, because of the difficulty of completely separating algal and fungal cells without lysing cells or losing significant biomass. To address this issue, we used fatty acid profiling of *N. oceanica* and *M. elongata* to identify a biomarker, linolenic acid (C18:3), which is a fatty acid that is predominantly present in the fungus (**Du et al., 2018a**). Thus, we used linolenic acid as a proxy to quantify the fungal biomass taking into account that the linolenic acid composition in the fungal biomass was consistent following the incubation in N-deprived f/2 medium (**Figure 3—figure supplement 2A**). Due to the tight interaction between the algae and fungi, it is impractical to accurately determine the correlation of the biomarker with fungal biomass under co-culturing conditions. Instead, we used the correlation of C18:3 with biomass of the fungus grown in N-depleted f/2 medium as a proxy for the fungal biomass in the co-cultures. We made the assumption that relative change of C18:3 in co-cultured and free cells were insignificant allowing for an accurate estimate of fungal biomass in both conditions. Linolenic acid was quantified by GC of its fatty acid methyl ester derivative, from which fungal biomass was calculated. The algal biomass was calculated by subtraction of fungal biomass from the total biomass of alga-fungus aggregates. Significant increases in biomass were observed for the co-cultured alga and fungus, but not when the alga or fungus were grown by themselves (**Figure 3—figure supplement 2B**). Therefore, both partners benefitted in this interaction. *M. elongata* was able to grow in nutrient-deprived conditions (PBS buffer) in the presence of the algal photobiont, but not when it was incubated by itself in PBS buffer without carbon (**Video 1**). Thus, both *N. oceanica* and *M. elongata* appear to benefit from their interaction and nutrient exchanges.

## Specificity of fungi hosts in interactions with *N. oceanica*

Numerous lineages of fungi have evolved to interact with plants and algae. The question arises whether the interaction we observed is unique to *Mortierella* or, alternatively, if it is conserved across diverse lineages of fungi. We addressed this through a series of interaction experiments pairing *N. oceanica* with a panel of 20 fungi (**Figure 3—figure supplement 3A**). These phylogenetically diverse fungal isolates represented three phyla, 9 orders and 13 families of fungi across trophic strategies from plant-associated fungal mutualists to pathogens and included the yeast *Saccharomyces cerevisiae*, as well as filamentous ascomycetes, basidiomycetes, and mucoromycetes (**Bonito et al., 2016**). *Mortierella elongata* showed the most obvious phenotype of flocculating alga, which consisted of algal cells clustered around the fungal mycelium (**Figure 3—figure supplement 3B**). Aside from a few *Mortierella* species tested, interactions between the other fungi and the alga were neutral or negative. It is worth noting that *N. oceanica* cells maintained an intact and smooth outer layer when co-cultured with the negatively interacting fungi such as *Clavulina* sp. PMI390 and *Morchella americana* GB760 (**Figure 3—figure supplement 4**).

## Long-term co-cultivation leads to internalization of *N. oceanica* within *M. elongata* hyphae

Microbial consortia may persist in a stable state, improving each other's resilience to fluctuating environments and stresses (**Brenner et al., 2008**). To assess whether the observed interaction between *N. oceanica* and *M. elongata* was stable or transient, we carried out a series of long-term incubations (from 1 to 3 months) in which the partners were grown together and nutrients refreshed biweekly. After ~1 month of co-culture, confocal microscopy was used to visualize cells inside the

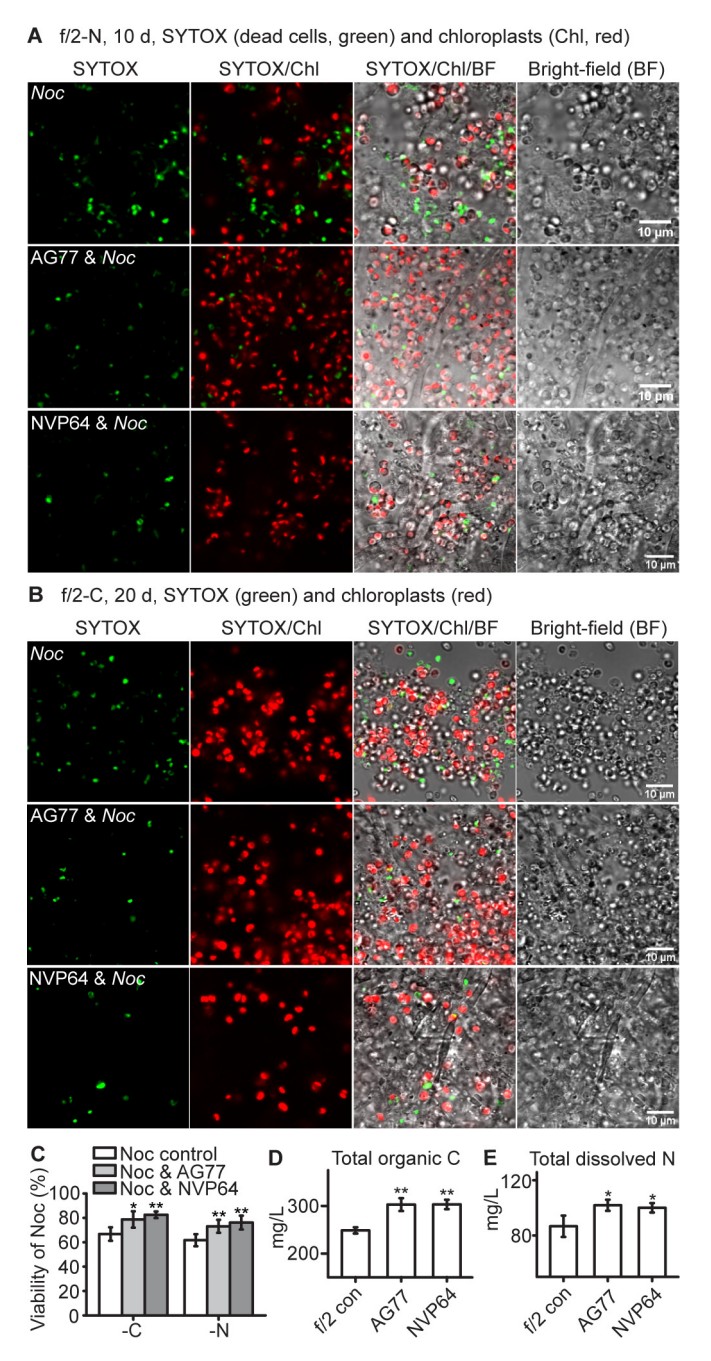

**Figure 3.** *N.oceanica* benefits from co-culture with *M. elongata*. (**A–C**) Viability assay of *Noc* cells and *Noc* co-cultured with AG77 under nitrogen (-N, (**A**) and carbon (-C, (**B**) deprivation. Dead *Noc* cells were indicated by SYTOX Green staining (green fluorescence). Red, *Noc* chlorophyll fluorescence. (**C**) Viability of nutrient-deprived *Noc* cells increased when co-cultured with two different *M. elongata* strains, AG77 and NVP64. Results are calculated from 1000 to 5000 cells of five biological repeats with ImageJ. Asterisks indicate significant differences compared to the *Noc* control as determined by Student's *t* test (*$p \leq 0.05$, **$p \leq 0.01$; Means $\pm$ SD, n = 5). (**D and E**) Total organic C and dissolved N measurements in the buffer of 18-day fungal cultures of *M. elongata* strains AG77 and NVP64 compared to the f/2 medium control (f/2 con). Fungal cells were removed by 0.22-μm filters. Data are presented as the average of four biological replicates and asterisks indicate significant differences compared to the f/2 medium control as determined by Student's *t* test. Means $\pm$ SD, n = 4. *$p \leq 0.05$, **$p \leq 0.01$.

DOI: https://doi.org/10.7554/eLife.47815.011

*Figure 3 continued on next page*

*Figure 3 continued*

The following figure supplements are available for figure 3:

**Figure supplement 1.** *N. oceanica* and *M. elongata* under stresses.
DOI: https://doi.org/10.7554/eLife.47815.012

**Figure supplement 2.** *N.oceanica* and *M. elongata* AG77 benefit from each other under nutrient starvation.
DOI: https://doi.org/10.7554/eLife.47815.013

**Figure supplement 3.** Screening of fungal isolates from diverse clades for intensive interaction with *N.oceanica*.
DOI: https://doi.org/10.7554/eLife.47815.014

**Figure supplement 4.** Scanning electron microscopy of co-cultures of *N.oceanica* with two fungal strains that did not trap *N. oceanica* cells.
DOI: https://doi.org/10.7554/eLife.47815.015

thick aggregates that formed between the alga and the fungus. To delineate cell walls, we used a wheat germ agglutinin conjugate cell wall probe, which binds to *N*-acetylglucosamine, a component in both the *Mortierella* and *Nannochloropsis* cell walls (*Javot et al., 2007*; *Scholz et al., 2014*). Microscopic observations indicated the presence of algal cells within fungal hyphae (*Figure 4—figure supplement 1A–C* and *Video 2*). Subsequent light and transmission electron microscopy (TEM) were used to further observe this phenomenon, whereby algal cells had been incorporated within hyphae. Differential interference contrast (DIC) microscopy showed the morphology of the 'green hyphae' after long-term co-culture, corroborating the presence of intact and presumably functional algal cells attached to the hyphal tip (*Figure 4A*) and present inside the fungal hyphal cells (*Figure 4B–E* and *Video 3*). After long-term co-culture, algae-fungi aggregates became thick and difficult to observe well with light microscopy (*Figure 4—figure supplement 1D and E*). The viability of *M. elongata* was demonstrated by transferring *M. elongata-N. oceanica* aggregates to fresh PDB/2 plates (*Figure 4—figure supplement 1F*). Additional imaging with TEM was performed to characterize the *M. elongata-N. oceanica* aggregates. Algal cells were seen outside of fungal cells surrounded by the fungal mycelium (*Figure 4—figure supplement 1G–I*); however, some algal cells are clearly present within the hyphae (*Figure 4F and G* and *Figure 4—figure supplement 2*). Fungal cytoplasmic contents were visible suggesting fungal cells containing algae were alive and functional (*Figure 4*).

While there is no indication that algae are transmitted vertically through fungal reproductive structures, the algal cells remained viable (growing and dividing) during 2 months of co-culture (*Video 4*). We were not able to capture the exact transitional stage of entry of *N. oceanica* into hyphae of *M. elongata* by TEM; however, through DIC and time-lapse microscopy, we repeatedly observed that internalization of algae is preceded by dense aggregation of algal cells around the hyphal tip (*Figure 4—figure supplement 3*). Dense clusters of algal cells at the tip of a hypha were consistently observed when algal cells were found within fungal hyphae growing in a semisolid medium (*Figure 4—figure supplement 4*). Furthermore, hyphae proximal from these tips were often green, and the number of algae within these cells increased over time (*Figure 4A–E*). In fact, trapped algal cells were able to grow and divide within their host (*Video 4*). To further examine the viability

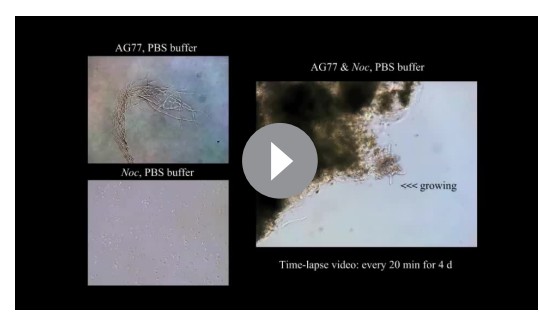

**Video 1.** *M. elongata* AG77 mycelia (~2 days in PDB), *N. oceanica* (*Noc*, 5 days in f/2) or AG77-*Noc* aggregates (7-d co-culture) were washed three times with phosphate-buffered saline (PBS, pH7.0–7.2, Life Technologies) and incubated in flasks containing PBS for ~2 days. Samples were then transferred to 35-mm-microwell dishes (glass top and bottom, MatTek) containing semisolid medium (PBS supplemented with 0.25% low-gelling-temperature agarose). The growth of samples was recorded by time-lapse photography (every 20 min for 4 days) using a Leica DMi8 inverted microscope with DIC and time-lapse function. Resultant images were used to create the movie with video-editing software (VideoStudio X9, Corel) to compare the growth of AG77 with or without symbiotic algal cells in nutrient-limited PBS buffer. Only hyphae of AG77-*Noc* aggregates kept growing, indicating that AG77 benefits from the co-culture with *Noc* cells.
DOI: https://doi.org/10.7554/eLife.47815.016

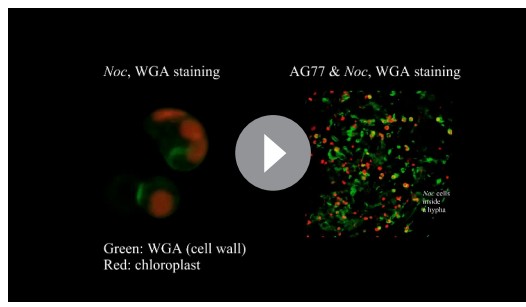

**Video 2.** Animation of 3D z-stacks of *N. oceanica* (*Noc*) or *M. elongata* AG77-*Noc* aggregates (35-day co-culture) stained by Wheat Germ Agglutinin Conjugate (WGA) and observed with a confocal laser scanning microscope (FluoView 1000, Olympus). Green, WGA fluorescence indicates cell wall of *Noc* and AG77; red, *Noc* chlorophyll fluorescence.
DOI: https://doi.org/10.7554/eLife.47815.024

of green hyphae, confocal microscopy with SYTOX Green was carried out in the 1 ~ 2 months alga-fungus aggregates. Exclusion of the dye in this case is primarily an indicator of living fungal hyphae, while persistent green chlorophyll and dividing cells are hallmarks of living algae. The results are consistent with the notion that both fungal host and internalized algae within the hyphae are alive (*Figure 4—figure supplement 5*). DIC microscopy also confirmed that the algal cells inside green hyphae are surrounded by fungal organelles, especially what appear to be lipid droplets (*Figure 4—figure supplement 6* and *Video 5*).

## Discussion

Here, we show that the alga *N. oceanica* and the fungus *M. elongata* have a strong and specific interaction, whereby *N. oceanica* aggregates along the fungal mycelium during co-incubation. We further demonstrate that in co-culture, living algal and fungal partners establish a tight physical association preceded by the loss of the smooth outer portion of the algal cell wall revealing fibrous extensions, and eventually leading to the incorporation of algal cells within the fungal mycelium. Isotope tracer studies provided evidence for the reciprocal exchange of carbon and nitrogen between living algal and fungal symbionts. Results from isotope tracer and nutrient deficiency studies support our hypothesis that the *Mortierella-Nannochloropsis* interaction is mutualistic in nature, based upon carbon and nitrogen acquisition and transfer, with potentially adaptive benefits provided to both partners under nutrient-limited conditions. While the apparent fungal-algal symbiosis may conjure the concept of a lichen, it differs in many respects. *Mortierella* lacks distinct tissue differentiation or hyphal structures (i.e. thallus, haustoria). Moreover, hyphae of *Mortierella* harbor algal cells intracellularly while lichens maintain algae in a fungal matrix, but external to their cells. In fact, we know of no other examples of fungi that have been shown to host intracellular eukaryotic photobionts.

Endosymbiosis of living eukaryotic cells by fungal hypha is not known from nature nor the lab. However, *Geosiphon pyriformis*, an early diverged fungal relative of *Mortierella* and arbuscular mycorrhizal fungi, does form a unique intracellular association with the photosynthetic cyanobacterium *Nostoc punctiforme* (*Mollenhauer et al., 1996*). In this fungal-bacterial photobiont symbiosis, the fungus envelops the photosynthetic *Nostoc* within a specialized swollen multinucleate fungal 'bladder' that is morphologically distinct from the rest of the fungal mycelium (*Schüßler et al., 1996*). Within this bladder, the cyanobacteria are surrounded by a host-derived symbiosome membrane specialized for acquiring photosynthate (*Brenner et al., 2008*).

Our attempts to capture the biogenesis of *N. oceanica* internalization within *M. elongata* through DIC and time-lapse microscopy show that algal internalization is preceded by dense aggregates of algal cells close to the fungal hyphal tip (*Figure 4—figure supplement 3A and B*). The hyphal tip is the actively growing region of the fungal colony, and a point of growth, membrane endocytosis, and cell wall construction (*Steinberg, 2007*). Aggregates of algal cells surrounding hyphal tips were also frequently observed in the long-term co-culture of the fungus and the alga (*Figure 4—figure supplement 3C–F*), and dense clusters of algal cells forming at the tip of a hypha were observed when partners were grown in a semisolid medium (*Figure 4—figure supplement 4*). Hyphae proximal from these tips were often green, and the number of algae within the cells increased over time (*Figure 4A–E*). Not only do algal cells enter the fungal hyphae, but their plastids were intact with all indications of being healthy and photosynthetically active. Further, *Nannochloropsis* cells continued to grow and multiply within the fungal mycelium (*Video 4*) and both fungal host and algae inside of green hyphae remained alive after 2 months co-culture (*Figure 4—figure supplements 5* and *6* and *Video 5*). The living fungal hyphae exclude SYTOX green invalidating this assay for the internalized algae as they may never see the dye. However, the algal cells inside the hypha showed

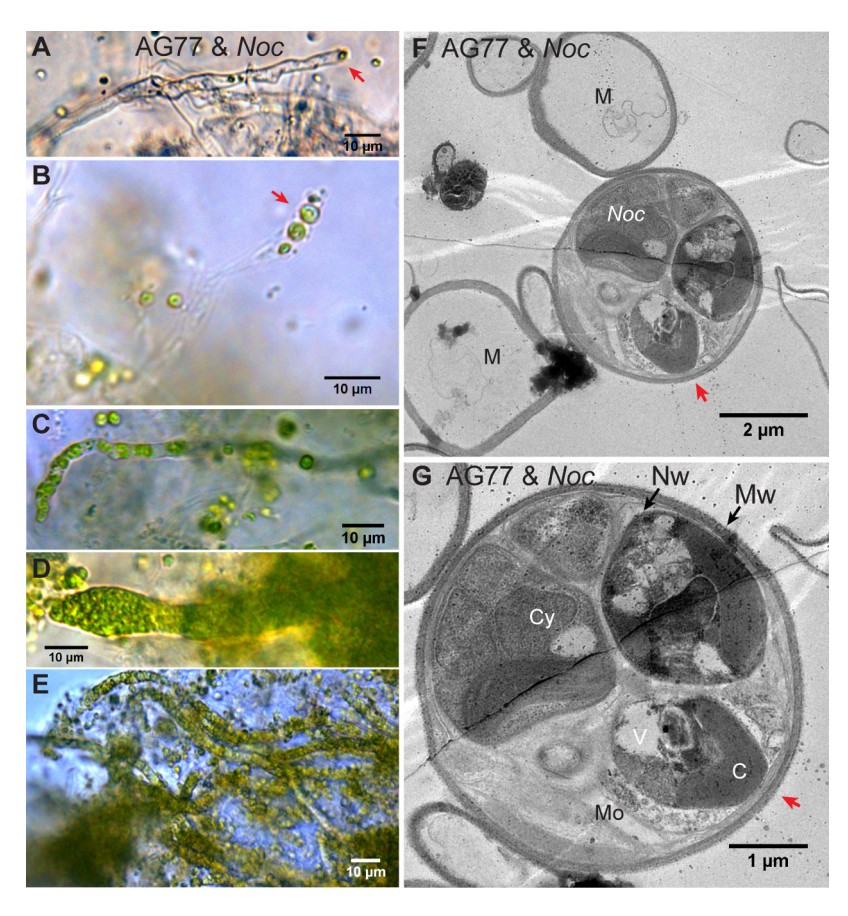

**Figure 4.** Intracellular localization of long-term co-cultured *N.oceanica* within *M. elongata* AG77 hyphae. (**A–E**) DIC images of AG77 'green hyphae' with *N. oceanica* (*Noc*) cells inside. The red arrow heads indicate *Noc* cells at the tip region of the hypha. (**B and C**) AG77 and *Noc* co-cultured for ~1 month. (**C–E**) AG77 and *Noc* co-cultured over 2 months. (**F and G**) Transmission electron microscope (TEM) images of increasing magnification showing a cross-section of AG77 mycelium containing a cluster of *Noc* cells. AG77 and *Noc* were co-cultured for ~1 month. Red arrowheads indicate the same position. M, mycelium; Nw, *Noc* cell wall; Mw, *Mortierella* cell wall; Cy, cytoplasm; V, vacuole C, chloroplast; Mo, *Mortierella* organelles.
DOI: https://doi.org/10.7554/eLife.47815.017

The following figure supplements are available for figure 4:

**Figure supplement 1.** Interaction between *N. oceanica* and *M. elongata* AG77 during long-term co-culture by confocal and transmission electron microscopy.
DOI: https://doi.org/10.7554/eLife.47815.018

**Figure supplement 2.** *N. oceanica* cells inside *M. elongata* mycelium after long-term co-culture.
DOI: https://doi.org/10.7554/eLife.47815.019

**Figure supplement 3.** Origin of *N.oceanica* within *M. elongata* AG77.
DOI: https://doi.org/10.7554/eLife.47815.020

**Figure supplement 4.** Presence of *N.oceanica* in *M. elongata* AG77.
DOI: https://doi.org/10.7554/eLife.47815.021

**Figure supplement 5.** Viability assay of green hyphae.
DOI: https://doi.org/10.7554/eLife.47815.022

**Figure supplement 6.** Light microscopy of green hyphae showing the coexistence of *N.oceanica* and fungal organelles inside hyphae.
DOI: https://doi.org/10.7554/eLife.47815.023

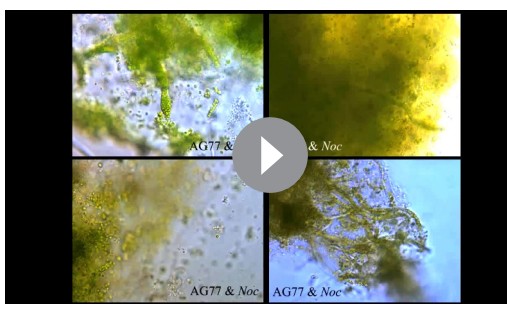

**Video 3.** Videos recorded with a Leica DMi8 inverted microscope to show the morphology of green hyphae in AG77-*Noc* aggregates (co-cultured over 2 months). DOI: https://doi.org/10.7554/eLife.47815.025

autofluorescence of chlorophyll, consistent with assembled photosynthetic membranes inside chloroplasts that are usually observed in living cells but not in dead disintegrating cells. In addition, it should be noted that instantly killed algal cells may still have chlorophyll autofluorescence as indicated by the white arrow heads in *Figure 4—figure supplement 5C*. Thus, a better indicator for living algal cells in the hyphae is the fact that they are actively dividing. Ultimately, we hypothesize that the hyphal tip is the initial point of entry for the algal cells into the hyphae, as the hyphal tips are the least differentiated cell features in a mycelial network, and the site of continuous plasma membrane recycling and cell wall generation.

Irrespective of whether the species studied here are ancient fungal-algal symbionts, or whether our findings demonstrate a latent capacity for intricate fungal-algal interactions, *Mortierella* fungi do share habitat and ecological niches with other Chromealveolate and algae. Although we have studied these organisms in a synthetic co-culture, given the global distribution of *M. elongata* and *N. oceanica*, it is plausible that these organisms (or their relatives) interact naturally, such as in marine tidal zones, estuaries or other ecotones.

Many endosymbionts are acquired laterally, such as *Symbiodinium* photobionts of coral (*Mies et al., 2017*). Other endosymbionts are heritable, such as the bacterial endosymbiont *Glomeribacter gigasporarum*, which is transmitted in spores of mycorrhizal fungi (*Bianciotto et al., 2004*). Given the relative size of *Nannochloropsis* cells (2–3 µm) to those of sporangiospores (2–3 µm), chlamydospores (23–30 µm) and zygospores (25–38 µm) of *M. elongata*, it is possible that internalized algae are transmitted vertically to fungal progeny. However, we have no evidence for heritability at this time.

Fungi are ubiquitous and function as root symbionts for a majority of land plant lineages and are regarded as having an essential role in the terrestrialization of Earth (*Field et al., 2016*). Recent phylogenomic analyses of Fungi resolve Mucoromycota as the earliest monophyletic lineage of plant-associated fungi (*Spatafora et al., 2016*), which include *Glomeromycotina* (arbuscular mycorrhizal fungi), *Mucoromycotina* (sugar-fungi), and *Mortierellomycotina* (soil molds). One trait that characterizes fungi in the Mucoromycota is their ability to form intimate cross-kingdom intracellular symbioses, which may be facilitated by the lack of regular septate cross walls between hyphal cells. These fungi form mycorrhizal symbiosis with extant early diverging plant lineages including hornworts, liverworts and mosses (*Redecker et al.,*

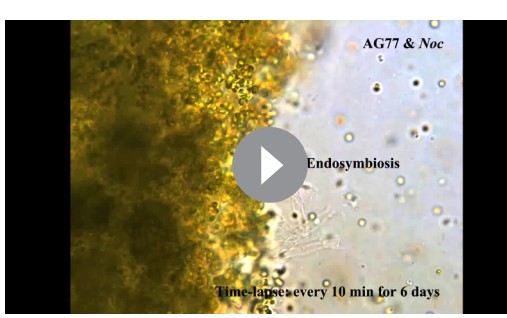

**Video 4.** *N. oceanica* (*Noc*) cells internalized within a *M. elongata* AG77 hypha recorded by time-lapse photography (every 10 min for 6 d), showing several *Noc* cells growing and dividing within the hypha. DOI: https://doi.org/10.7554/eLife.47815.026

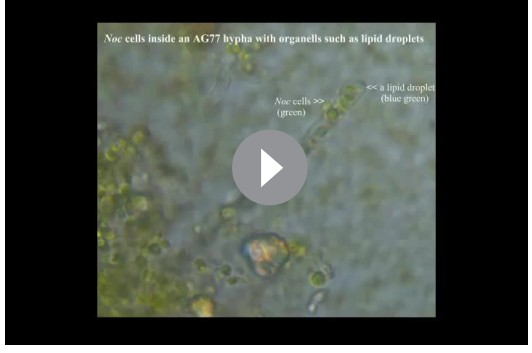

**Video 5.** Light microscope video to show the *N. oceanica* (*Noc*) cells inside a *M. elongata* AG77 hypha. The green *Noc* cells are surrounded by fungal organelles such as lipid droplets (blue green/gray) that are presented in living hypha. DOI: https://doi.org/10.7554/eLife.47815.027

*2000*; *Spatafora et al., 2016*). This radiation of Mucoromycota involved the loss of flagella in this lineage of Fungi and was contemporaneous with at least two radiations of green algae prior to the emergence of terrestrial embryophytes (*Lutzoni et al., 2018*). We show here that *Mortierella* that belong to this lineage also interact with the microalgae *N. oceanica*. This finding enables future studies on cross-kingdom signaling, and genetic and metabolic factors underlying this symbiosis.

*Mortierella* ecology has evaded mycologists for centuries. These fungi are commonly isolated from soils and plant roots (*Summerbell, 2005*). They have also been isolated from strata directly under green macroalgae in Antarctica (*Furbino et al., 2014*). *Mortierella* is common within cryptobiotic desert crusts (along with bacteria, algae, and other fungi) (*Bates et al., 2010*), and *M. elongata* has even been detected in association with red algae in alpine snow packs (*Brown and Jumpponen, 2014*). Although commonly regarded as soil saprotrophs in the literature, our results demonstrate that at least some of these fungi are also involved in biotrophic mutualisms. Although *N. oceanica* is not a charophyte, the closest algal relative to land plants identified to date, our study shows that an early diverging fungus is adapted to form a biotrophic mutualism with Chromalveolate alga, indicating that algal-fungal mutualisms may be more ancient and diverse than previously recognized.

## Conclusions

Through stable- and radio-isotope-tracer experiments, metabolic analysis and microscopy, we report that the globally distributed early-diverging terrestrial fungus *M. elongata* interacts intimately with the marine alga *N. oceanica* in a mutualism that leads to the incorporation of intact living algal cells within fungal hyphae. This symbiosis appears to be based upon an exchange of carbon and nitrogen between the cells. *M. elongata* is the first taxon in the Kingdom Fungi that has been shown to internalize actively photosynthesizing eukaryotic cells.

# Materials and methods

**Key resources table**

| Reagent type (species) or resource | Designation | Source or reference | Identifiers | Additional information |
|---|---|---|---|---|
| Strain (*Nannochloropsis oceanica* CCMP1779) | *Noc* | Provasoli-Guillard National Center for Culture of Marine Phytoplankton | CCAP211/46 | Kuwait Institute for Scientific Research |
| Strain (*Mortierella elongata* AG77) | AG77/*Mel* AG77 | *Uehling et al., 2017* | | North Carolina |
| Strain (*Mortierella elongata* NVP64) | NVP64 | *Uehling et al., 2017* | | Michigan |
| Strain (*Mycoavidus cysteinexigens*) | *M. cysteinexigens* | *Uehling et al., 2017* | | |
| Commercial assay or kit | SYTOX Green | Thermo Fisher Scientific | R37168 | |
| Commercial assay or kit | BODIPY 493/503 | Thermo Fisher Scientific | D3922 | |
| Commercial assay or kit | Wheat Germ Agglutinin Conjugate Alexa Fluor 488 | Thermo Fisher Scientific | W11261 | |
| Commercial assay or kit | resin Epon/ Araldite mixture | Electron Microscopy Sciences | 13940 | |
| Chemical compound, drug | [14C]-sodium bicarbonate | American Radiolabeled Chemicals | ARC 0138 C-1 mCi | |
| Chemical compound, drug | [14C]-D-glucose | Moravek Biochemicals | MC144W | |
| Chemical compound, drug | [14C]-sodium acetate | American Radiolabeled Chemicals | ARC 0101A | |

*Continued on next page*

*Continued*

| Reagent type (species) or resource | Designation | Source or reference | Identifiers | Additional information |
|---|---|---|---|---|
| Chemical compound, drug | [15N]-ammonium chloride | Sigma-Aldrich | 299251 | |
| Software, algorithm | VideoStudio X9 | VideoStudio | X9 | |

## Strains and growth conditions

The marine alga *Nannochloropsis oceanica* CCMP1779 was obtained from the Provasoli-Guillard National Center for Culture of Marine Phytoplankton and incubated as previously described (*Vieler et al., 2012*). In brief, *N. oceanica* cells were grown in f/2 medium containing 2.5 mM $NaNO_3$, 0.036 mM $NaH_2PO_4$, 0.106 mM $Na_2SiO_3$, 0.012 mM $FeCl_3$, 0.012 mM $Na_2EDTA$, 0.039 μM $CuSO_4$, 0.026 μM $Na_2MoO_4$, 0.077 μM $ZnSO_4$, 0.042 μM $CoCl_2$, 0.91 μM $MnCl_2$, 0.3 μM thiamine HCl/vitamin B1, 2.05 nM biotin, 0.37 nM cyanocobalamin/vitamin B12, and 20 mM sodium bicarbonate and 15 mM Tris buffer (pH 7.6) to prevent carbon limitation. The cultures were incubated in flasks under continuous light (~80 μmol/m$^2$/s) at 22°C with agitation (100 rpm). Log-phase algal cultures (1 ~ 3×10$^7$ cells/mL) were used for co-cultivation with fungi. Cell size and density of algal cultures were determined using a Z2 Coulter Counter (Beckman). *Mortierella elongata* AG77 and NVP64 isolates were made from soil samples collected in North Carolina (AG77) and Michigan (NVP64), USA. *M. elongata* AG77 and NVP64 are known to contain an endosymbiotic bacterium, *Mycoavidus cysteinexigens*, and were cleared of this endosymbiont through a series of antibiotic treatments as previously described (*Partida-Martinez and Hertweck, 2007*; *Uehling et al., 2017*). The resultant *Mycoavidus*-free strains were used for the co-cultivation with *N. oceanica*. Other fungal strains used in this study were obtained from the fungal culture suppliers and isolated from sporocarps, soils, and from healthy surface-sterilized *Populus* roots obtained from the Plant-Microbial Interfaces project (*Bonito et al., 2016*). Fungi were incubated in flasks containing PDB medium (12 g/L potato dextrose broth and 5 g/L yeast extract, pH5.3) at room temperature (RT, ~22°C).

For co-culturing *N. oceanica* and fungi, fungal mycelia were briefly blended into small pieces (0.5 to 2 cm) using a sterilized blender. After a 24-hr recovery in PDB medium, fungal cells were collected by centrifugation (3000 *g* for 3 min), washed twice with f/2 medium and resuspended in ~15 mL f/2 medium. A portion of fungal mass (3–4 mL) was used for the calculation of dry biomass: 1 mL was transferred and filtered through pre-dried and pre-weighed Whatman GF/C filters and dried overnight at 80°C. A similar method was used for the measurement of algal biomass. About a 3:1 ratio of fungal:algal biomass was used for co-cultivation on a shaker (~60 rpm) under continuous light (~80 μmol/m$^2$/s) at RT. After 18-day co-culture, the shaker was turned off to allow free settling of the algal and fungal cells overnight. The supernatant was removed and the same volume of fresh f/2 medium containing 10% PDB was added to the culture. After that, the alga-fungus co-culture was refreshed biweekly with f/2 medium supplemented with 10% PDB.

Nutrient deprivation of the co-culture was performed according to a published protocol for *N. oceanica* (*Vieler et al., 2012*). Mid-log-phase *N. oceanica* cells (~1×10$^7$ cells/mL) grown in f/2 media (25 mL) were harvested by centrifugation and washed twice with nutrient-deficient f/2 media [without carbon (-C), nitrogen (-N) or phosphorus (-P)] and resuspended in 25 mL nutrient-deficient f/2 media, respectively. AG77 mycelia grown in PDB medium were washed twice with the nutrient-deficient f/2 and added into respective *N. oceanica* cultures for co-cultivation. To block air exchange, the flasks of -C cultures were carefully sealed with Parafilm M over aluminum foil wrap. Cell viabilities were analyzed by confocal microscopy after 10-day co-culture of -N and 20 days of -C.

## Light microscopy

Interaction and symbiosis between the alga and the fungus were examined with an inverted microscope with differential interference contrast (DIC) and time-lapse modules (DMi8, Leica). DIC images were taken from the alga-fungus aggregates after short- (6 days) and long-term (over 1 month) co-cultivation. To characterize the algal endosymbiosis in the fungus, DIC and time-lapse photography were performed after long-term co-culture of the alga and fungus (from 1 to 3 months). For viewing

alga-fungus aggregates grown in flasks, the samples were transferred to 35-mm-microwell dishes (glass top and bottom, MatTek) and embedded in a thin layer of semisolid f/2 medium supplemented with 10% PDB and 0.25% low-gelling-temperature agarose (Sigma-Aldrich) to immobilize the cells. The morphology of green hyphae (AG77 hyphae containing intracellular *N. oceanica* cells) was recorded in DIC micrographs, as well as real-time videos that showed four groups of green hyphae (*Video 3*). Videos were assembled side by side in *Video 3* using video-editing software VideoStudio X9 (Corel). To investigate the establishment of algal cells living inside fungal hypha, randomly selected alga-fungus aggregates were sub-cultured from 35-day co-cultures in 35-mm-microwell dishes containing semisolid f/2 medium with 10% PDB and 0.25% agarose and observed directly in 35-mm-microwell dishes containing semisolid f/2 medium (*Figure 4—figure supplement 3*; *Figure 4—figure supplement 4*) and through time-lapse photographs that were combined together with the software VideoStudio to create *Video 4*.

## Scanning electron microscopy

SEM was performed at the Center for Advanced Microscopy of Michigan State University (CAM, MSU) to investigate the physical interaction between *N. oceanica* and *M. elongata*. Alga-fungus aggregates from 6-day co-cultures of *N. oceanica* and fungal strains were used for interaction analysis, including *M. elongata* AG77, NVP64 and *Clavulina* PMI390 and *Morchella Americana* GB760, which do not have interaction phenotype when co-cultured with algae. *N. oceanica* cells grown alone in f/2 medium were used as a control. We also observed *N. oceanica* cells co-cultured with 65°C-killed AG77 mycelium, and algal cells from the supernatant of living *M. elongata-N. oceanica* co-cultures that were unattached from fungal-algal aggregates. To mimic the exposed fibrous extensions of *N. oceanica* cells following physical interaction with *M. elongata*, different enzymes were tested to digest the out layer of algal cell wall. *N. oceanica* cells were washed with PBS buffer and incubated with different combination of enzymes in PBS buffer at RT for 3 hr: 4% hemicellulase (mixture of glycolytic enzymes such as xylanase and mananase, Sigma-Aldrich); 2% driselase (mixture of carbohydrolases including laminarinase, xylanase, and cellulase, Sigma-Aldrich); 4% hemicellulase and 2% driselase; 1% chitinase (Sigma-Aldrich); 1% lysing enzymes (mixture of glucanase, protease, and chitinase, Sigma-Aldrich). The samples were fixed in 4% (v/v) glutaraldehyde solution and dried in a critical point dryer (Model 010, Balzers Union). After drying, the samples were mounted on aluminum stubs using high vacuum carbon tabs (SPI Supplies) and coated with osmium using a NEOC-AT osmium coater (Meiwafosis). Processed tissues were examined using a JSM-7500F scanning electron microscope (Japan Electron Optics Laboratories).

## Nutrient exchange

Light microscopy and SEM showed a close physical interaction between *N. oceanica* and *M. elongata* that led us to examine whether there is metabolite exchange between *N. oceanica* and *M. elongata* by isotope labeling and chasing experiments with carbon and nitrogen ($^{14}$C and $^{15}$N), two of the most important nutrients for *N. oceanica* and *M. elongata*. $^{14}$C assays were performed according to published protocols with modifications (*Li et al., 2012*). 20 μL of [$^{14}$C]-sodium bicarbonate (1 mCi/mL, 56 mCi/mmol, American Radiolabeled Chemicals) was added to 20 mL of early log-phase culture of *N. oceanica* (~$2 \times 10^6$ cells/mL) and incubated for 5 days when the $^{14}$C incorporation reached ~40%. The $^{14}$C-labeled *N. oceanica* cells were harvested by centrifugation (4000 *g* for 10 min) and washed three times with f/2 medium. The supernatant of the last wash was analyzed in Bio-Safe II counting cocktail (Research Products International) using a scintillation counter (PerkinElmer 1450 Microbeta Trilux LSC), to confirm that $^{14}$C-labeling medium was washed off. The pellet of $^{14}$C-labeled *N. oceanica* was resuspended in 20 mL f/2 medium. Subsequently, non-labeled *M. elongata* AG77 mycelia (~3 x algal biomass, intact cells without blending) grown in PDB medium were washed twice with f/2 medium and added to the 20 mL $^{14}$C-labeled algal culture for 7-day co-cultivation. Alga-fungus aggregates were then harvested by PW200-48 mesh (Accu-Mesh, first filtration) and NITEX 03-25/14 mesh (mesh opening 25 μm, SEFAR, second filtration). Algal cells in the flow through were collected by centrifugation (4000 *g* for 10 min) and kept as the first part of $^{14}$C-labeled alga control. Alga-fungus aggregates were intensively washed in 50-mL conical centrifuge tubes containing 40 mL of f/2 medium using a bench vortex mixer (~1500 rpm, 15 min). Fungal mycelia were collected with NITEX 03-25/14 mesh; algal cells in the flow through were harvested by centrifugation

and stored as the second fraction of $^{14}$C-labeled alga control. Mesh-harvested fungal mycelia (with obviously reduced the number of algal cells attached) were placed in microcentrifuge tubes containing 300 µL of PBS buffer (pH 5.0) supplemented with 4% hemicellulase and 2% driselase for overnight incubation at 37°C to digest the algal cell walls as previously described (*Chen et al., 2008*). After cell-wall digestion, 700 µL of f/2 medium were added and the algal cells were separated from hyphae by vortexing for 15 min. The hyphae were collected by NITEX 03-25/14 mesh, and the flow-through containing algal cells was kept as the last fraction of alga control. The fungal hyphae were washed three times with f/2 medium and then used for biomass and radioactivity measurements. The three fractions of $^{14}$C-labeled alga controls were combined together for further analyses. Half of the algal and fungal samples were dried and weighed for biomass and the rest was used for $^{14}$C measurements. The $^{14}$C radioactivity of each sample was normalized to the respective dry biomass. To examine cross contamination after alga-fungus isolation, non-radioactive samples were processed the same way and analyzed by light microscopy (*Figure 2—figure supplement 1A–C*) and PCR using primers specific for the *N. oceanica* gene encoding Aureochrome 4 (*AUREO4*), a blue light-responsive transcription factor that is unique in photosynthetic stramenopiles such as *N. oceanica* (*Figure 2—figure supplement 1D*): Aureo4pro F+ (5′-AGAGGAGCCATGGTAGGAC-3′) and Aureo4 DNAD R- (5′-TCGTTCCACGCGCTGGG-3′), and primers specific for *M. elongata* genes encoding translation elongation factor EF1α and RNA polymerase RPB1 (*Figure 2—figure supplement 1E*): *EF1αF* (5′-CTTGCCACCCTTGCCATCG-3′) and *EF1αR* (5′-AACGTCGTCGTTATCGGACAC-3′), *RPB1F* (5′-TCACGWCCTCCCATGGCGT-3′) and *RPB1R* (5′-AAGGAGGGTCGTCTTCGTGG-3′).

Isolated algal and fungal cells were frozen in liquid nitrogen and ground into fine powders with steel beads and TissueLyser II (QIAGEN), followed by lipid extraction in 1.2 mL chloroform:methanol (2:1, v/v) by vortexing for 20 min. After addition of double-distilled water (ddH$_2$O, 100 µL), the samples were briefly vortexed and then centrifuged at 15,000 *g* for 10 min. The organic phase was collected for total lipids. One mL of 80% methanol (v/v) was added to the water phase and cell lysis to extract free amino acids (FAAs). After centrifugation at 20,000 *g* for 5 min, the supernatant was kept as total FAAs and the pellet was air-dried; 200 µL of SDS buffer (200 mM Tris-HCl, 250 mM NaCl, 25 mM EDTA, 1% SDS, pH7.5) was added to the pellet with incubation at 42°C for 15 min. After centrifugation at 10,000 *g* for 10 min, while the pellet was kept for carbohydrate analyses, the supernatant (~200 µL) was collected for further protein precipitation (−20°C, 1 hr) with the addition of 800 µL cold acetone. After the 1 hr precipitation, total proteins (pellet) and soluble compounds (supernatant) were separated by centrifugation at 20,000 *g* for 15 min. The pellet of total proteins was resuspended in 200 µL of SDS buffer for scintillation counting. The pellet of carbohydrates was air-dried, resuspended in 200 µL ethanol, transferred to a glass tube with Teflon-liner screw cap, and then dissolved in 2 to 4 mL of 60% sulfuric acid (v/v) according to described protocols (*Velichkov, 1992*; *Scholz et al., 2014*). As needed, vortexing and incubation at 50°C were performed. Total lipids and soluble compounds were counted in 3 mL of xylene-based 4a20 counting cocktail (Research Products International), whereas total FAAs, proteins and carbohydrates were counted in 3 mL of Bio-Safe II counting cocktail. $^{14}$C radioactivity of the samples (dpm, radioactive disintegrations per minute) was normalized to their dry weight (dpm/mg).

To examine carbon transfer from the fungus to the alga, 200 µL of 0.1 mCi/mL [$^{14}$C]-D-glucose (268 mCi/mmol, Moravek Biochemicals) or 100 µL of 1 mCi/mL [$^{14}$C]-sodium acetate (55 mCi/mmol, American Radiolabeled Chemicals) were added to 20 mL of *M. elongata* AG77 grown in modified Melin-Norkrans medium [MMN, 2.5 g/L D-glucose, 0.25 g/L (NH4)$_2$HPO4, 0.5 g/L KH$_2$PO4, 0.15 g/L MgSO4, 0.05 g/L CaCl$_2$]. After 5-d $^{14}$C-labeling, fungal mycelia were harvested and washed three times with f/2 medium. The supernatant of the last wash was confirmed to be free of $^{14}$C by scintillation counting. $^{14}$C-labeled hyphae were added to 20 mL of *N. oceanica* culture for 7-day co-culture. Alga-fungus aggregates were harvested using PW200-48 and NITEX 03-25/14 meshes. Algal cells in the flow-through were harvested and washed twice with f/2 medium by centrifugation and kept as free *N. oceanica* (unbound algal cells). The remaining steps of sample preparation and $^{14}$C measurement were performed as described above.

To test whether physical contact is necessary for the carbon exchange between *N. oceanica* and *M. elongata*, $^{14}$C-label experiments were carried out using standard six-well cell culture plates (5 mL medium of each well) with inserts that have a bottom composed of hydrophilic polytetrafluoroethylene membrane filters (pore size of 0.4 µm, Millipore) to grow the alga and fungus together, which allows metabolite exchange but no physical contact. $^{14}$C-labeling was performed in the same way as

described above. For alga-fungus co-culture, $^{14}$C-labeled algal cells (or fungal hyphae) were added in either plate wells or cell culture inserts while respective hyphae (or algal cells) were grown separately in the inserts or plate wells to examine cross contamination (*Figure 2—figure supplement 2A*). After 7-day co-culture, algal and fungal cells grown in the insert-plate system were easily separated by moving the insert to an adjacent clean well (*Figure 2—figure supplement 2B and C*). Samples were then processed following the protocol described above (without the steps of mesh filtration and cell-wall digestion).

Considering that *Mortierella* fungi are saprotrophic (*Phillips et al., 2014*), we performed $^{14}$C-label experiments using heat-killed $^{14}$C-cells to test whether the alga and fungus utilize $^{14}$C from dead cells. Briefly, $^{14}$C-labeled algal or fungal cells were washed three times with f/2 medium and incubated in a water bath at 65°C for 15 min, which killed the cells without causing significant cell lyses. Heat-killed $^{14}$C-algal cells (or fungal hyphae) were co-cultivated with unlabeled hyphae (or algal cells) for 7 days in flasks. Subsequently, the algal and fungal cells were separated by cell-wall digestion and mesh filtration, and $^{14}$C radioactivity of the samples was measured by scintillation counting as described above.

Nitrogen is another major nutrient for *N. oceanica* (*Vieler et al., 2012*; *Zienkiewicz et al., 2016*) and *Mortierella* (*Thornton, 1956*). Nitrogen exchange between *N. oceanica* and *M. elongata* was tested by $^{15}$N-labeling and chasing experiments using isotope ratio mass spectrometry. For $^{15}$N labeling of algal and fungal cells, *N. oceanica* cells were inoculated and grown in 200 mL of $^{15}$N-f/2 medium containing ~5% of [$^{15}$N]-potassium nitrate [$^{15}$N/($^{15}$N+$^{14}$N), mol/mol], while *M. elongata* mycelia were inoculated and incubated in 2 L of $^{15}$N-MMN medium containing ~5% of [$^{15}$N]-ammonium chloride for two weeks. The algal culture was maintained in log phase by the addition of fresh $^{15}$N-f/2 medium into a larger volume. Eventually, $^{15}$N-*N. oceanica* cells from a 4 L culture and $^{15}$N-*M. elongata* mycelium from a 2-L culture were harvested by centrifugation, with a portion of each sample kept as $^{15}$N-labeled controls. The remainder of the samples was added to unlabeled cells in flasks (with physical contact) or 6-well-culture plates with inserts (no physical contact) for 7-day co-cultivation. Algal and fungal cells were separated after the co-culture as described above. Samples were washed three times with ddH$_2$O. Fungal mycelia were homogenized in a TissueLyser II (QIAGEN) using steel beads. The algal and fungal samples were then acidified with 1.5 to 3 mL of 1 N HCl, dried in beakers at 37°C and weighed for biomass. Isotopic composition of the samples [Atom % $^{15}$N, $^{15}$N/($^{15}$N+$^{14}$N)100%] and N content (%N) were determined using a Eurovector (EuroEA3000) elemental analyzer interfaced with an Elementar Isoprime mass spectrometer following a standard protocol (*Fry, 2007*). The N uptake rates (μmol N/mg biomass/d) of $^{15}$N-*N. oceanica* cells from the medium (medium-N, isotope dilution) and that of AG77 from $^{15}$N-*N. oceanica*-derived N ($^{15}$N) were calculated based on the Atom% $^{15}$N, %N and biomass following a published protocol (*Ostrom et al., 2016*). The N uptake rates of $^{15}$N-AG77 from the medium and that of recipient *N. oceanica* from $^{15}$N-AG77-derived N ($^{15}$N) were calculated in the same way.

## Confocal microscopy

Viability of *N. oceanica* and *M. elongata* cells during their co-culture was determined by confocal microscopy using a confocal laser scanning microscope (FluoView 1000, Olympus) at CAM, MSU. SYTOX Green nucleic acid stain (Molecular Probes, Thermo Fisher Scientific), a green-fluorescent nuclear and chromosome counterstain impermeant to live cells, was used to indicate dead cells following a published protocol (*Tsai et al., 2014*). Briefly, 1 μL of 5 mM SYTOX Green was added to 1 mL of cell culture and incubated for 5 min at RT in the dark. Samples were washed twice with f/2 medium before observation (SYTOX Green, 488 nm excitation, 510 to 530 nm emission; chlorophyll, 559 nm excitation, 655 to 755 nm emission). Viability of *N. oceanica* cells was calculated using ImageJ software. Cell viability was analyzed during the alga-fungus co-culture in flasks containing f/2 medium (1, 4 and 7 days, *Figure 2—figure supplement 3F–I*) to investigate whether the cells were alive or dead during the 7-day co-culture of $^{14}$C- and $^{15}$N-labeling experiments. Viability of *N. oceanica* cells co-cultivated with *M. elongata* AG77 and NVP64 under nutrient deprivation (-N and -C) was tested to show whether *N. oceanica* benefits from the co-culture with *Mortierella* fungi. Viability of *M. elongata* AG77 was analyzed during the 30-day incubation in f/2 medium (9, 18 and 30 days) to check whether the cells were alive or dead (*Figure 3—figure supplement 1*) when the culture media were collected using 0.22 μm Millipore filters after 18-day incubation for nutrient analyses

(total organic C and dissolved N). Viability of green hyphae containing algal cells was analyzed in the randomly selected alga-fungus aggregates after 1–2 months co-culture.

Localization of *N. oceanica* cells in alga-fungus aggregates was investigated by cell-wall staining using Wheat Germ Agglutinin Conjugate Alexa Fluor 488 (WGA, Thermo Fisher Scientific) following the manufacturer's instruction (*Figure 4—figure supplement 1A–C*). In brief, alga-fungus aggregates were collected by centrifugation and washed once with PBS buffer (pH7.2), followed by addition of 5 µg/mL WGA and incubation at 37℃ for 10 min. Samples were washed twice with f/2 medium and observed under a FluoView 1000 microscope (WGA, 488 nm excitation, 510 to 530 nm emission; chlorophyll, 559 nm excitation, 655 to 755 nm emission).

*M. elongata* AG77 hyphae contain many lipid droplets visible by light microscopy. To confirm the distribution of lipid droplets in hyphae grown in single and co-culture, confocal microscopy was carried out using BODIPY 493/503 (Thermo Fisher Scientific), a lipophilic probe for lipid droplets (488 nm excitation, 510 to 530 nm emission).

## Carbon and nitrogen measurements

Total organic C (TOC) and dissolved N (TDN) in the media of *Mortierella* cultures were measured with a TOC-Vcph carbon analyzer with total nitrogen module (TNM-1) and ASI-V autosampler (Shimadzu) at Kellogg Biological Station, MSU. *M. elongata* strains AG77 and NVP64 were incubated for 18 days in flasks containing 25 mL of f/2 medium. Fungal tissues were removed by filtration with 0.22 micron filters (Millipore) and the flow-through was subject to TOC and TDN analyses following published protocols (*Heinlein, 2013*; *Lennon et al., 2013*).

## Chlorophyll assay

Chlorophyll measurement was performed as previously described (*Du et al., 2018b*). In brief, *N. oceanica* cells were incubated in f/2 medium until they reached stationary-phase (0 day control), and the cells were further incubated for 10 days within the same medium or with the addition of about three-times biomass of f/2-washed and blot-dried AG77 mycelium. Algal cells were collected from 1 ml culture of *N. oceanica* controls and unbound cells from alga-fungus co-culture by centrifugation. Chlorophyll of the algal cells was extracted by 900 µl of acetone:DMSO (3:2, v/v) for 20 min with agitation at RT, and then measured with a spectrophotometer (Uvikon 930, Kontron).

## Fatty acid analysis and biomass calculation

Lipid extraction and fatty acid analysis were performed following a published protocol (*Du et al., 2018a*). Linolenic acid (C18:3) was used as a biomarker, as it is present in *M. elongata* AG77 but not in *N. oceanica* cells and its abundance in total biomass was steady following the incubation in N-deprived f/2 medium. Thus, C18:3 was quantified by gas chromatography of its methyl ester derivative and used for the calculation of fungal biomass in dense alga-fungus aggregates, when it was not feasible to physically separate algal and fungal cells without significant loss of biomass or cellular lysis. Briefly, alga-fungus aggregates were collected with mesh filtration and total lipid was extracted with methanol/chloroform/88% formic acid (1:2:0.1 by volume) and washed with 0.5 vol of 1 M KCl and 0.2 M $H_3PO_4$. After phase separation by centrifugation (3000 *g* for 3 min), total lipids were collected for the preparation of fatty acid methyl esters by transesterification and analysis by gas chromatography. The remaining cell lysate were dried at 80℃ overnight to provide the nonlipid biomass. Total dry biomass of alga-fungus aggregates was obtained by combining the lipid and non-lipid parts. Fungal biomass within alga-fungus aggregates was quantified using the C18:3 content-based calculation. Algal biomass in aggregates was determined by subtracting fungal biomass from the total biomass.

## Phylogeny of fungal strains

DNA was extracted from fungal isolates by placing a small amount of mycelium into 20 µL of extraction solution (Sigma-Aldrich) and heating at 95℃ for 10 min, after which 60 µL of bovine serum albumin (BSA, 3%) was added to the lysate and PCR was employed to directly amplify the nuclear-encoded ribosomal RNA genes (rDNA): ITS (internal transcribed spacer) with the primers ITS1f (5'-CTTGGTCATTTAGAGGAAGTAA-3') and ITS4 (5'-TCCTCCGCTTATTGATATGC-3'), and 28S rDNA with primers LROR (5'-ACCCGCTGAACTTAAGC-3') and LR3 (5'-CCGTGTTTCAAGACGGG-3')

following a published PCR protocol (*Bonito et al., 2016*). Amplicons were sequenced with an ABI3730XL automated sequencer (Applied Biosystems). The resultant sequences were identified by BLAST in the NCBI nucleotide database (*Altschul et al., 1990*), and by sequence alignment in MUS-CLE (*Edgar, 2004*). Unalignable regions were excluded in Mesquite (*Maddison and Maddison, 2009*). Phylogenetic relationships among isolates were inferred with PAUP* (*Swofford, 2002*) using the neighbor joining optimization criterion and were visualized with FigTree (*Rambaut, 2007*). The alignment of sequences used in this study has been deposited in TreeBase (#20243).

### Transmission electron microscopy.

TEM was performed at CAM, MSU using *N. oceanica* and *Mortierella* aggregates co-cultured for ~1 month. Randomly collected alga-fungus aggregates were fixed overnight at 4°C in sodium cacodylate buffer (50 mM, pH 7.2) supplemented with 2.5% (v/v) glutaraldehyde. The fixed samples were washed three times with sodium cacodylate buffer, post-fixed in 1% $OsO_4$ (v/v) for 2 hr at RT and then washed three times with sodium cacodylate buffer. After dehydration with a graded series of ethanol and acetone, the samples were infiltrated through a series of acetone/resin Epon/Araldite mixtures and finally embedded in resin Epon/Araldite mixture (Electron Microscopy Sciences). Ultra-thin sections (70 nm) were cut with an ultramicrotome (RMC Boeckeler) and mounted onto 150 mesh formvar-coated copper grids, followed by staining with uranyl acetate for 30 min at RT. The sections were then washed with ultrapure water and stained 10 min with lead citrate and used for observation. Images were taken with a JEOL100 CXII instrument (Japan Electron Optics Laboratories) with SC1000 camera (Model 832, Gatan) and were processed with ImageJ.

## Acknowledgements

We thank T James, J Tiedje, S Y He, B Sears, R Roberson and F Trail for critical discussion and review of this manuscript. We acknowledge J Uehling (Duke) and A Desirò (MSU) for detection and clearing of endobacteria from the fungal isolates used in this research and R Vilgalys (Duke) for use of the genome reference isolate of *Mortierella elongata* AG77. We thank A Albin, C Flegler, A Withrow, M Frame at the MSU Center for Advanced Microscopy for technical assistance of microscopy. We thank D Weed and S Hamilton (Kellogg Biological Station, MSU) for carbon and nitrogen analyses. We thank H Gandhi (Department of Integrative Biology, MSU) for technical assistance of $^{15}N$ analyses. We thank J Alvaro (Hope College, Michigan) for assistance in the preliminary test of interaction between algae and fungi. We thank D Schnell (MSU) for providing $[^{14}C]$-sodium bicarbonate and E Poliner and E Farre for providing *Aureo4* primers. This work was supported in part by a grant from the Chemical Sciences, Geosciences, and Biosciences Division, Office of Basic Energy Sciences, Office of Science, U.S. Department of Energy (DE-FG02-91ER20021) to CB. This material is based upon work supported by the Great Lakes Bioenergy Research Center, U.S. Department of Energy, Office of Science, Office of Biological and Environmental Research under Award Numbers DE-SC0018409 and DE-FC02-07ER64494. KZ received funding from the People Programme (Marie Curie Actions) of the European Union's Seventh Framework Programme FP7/2007−2013/under REA grant agreement n° [627266] supporting KZ. The EU is not liable for any use that may be made of the information contained therein. GB and CB are grateful to AgBioResearch for financial support.

## Additional information

### Funding

| Funder | Grant reference number | Author |
| --- | --- | --- |
| U.S. Department of Energy | DE-FG02-91ER20021 | Christoph Benning Zhi-Yan Du |
| U.S. Department of Energy | DE-SC0018409 | Nathaniel E Ostrom Gregory M Bonito |
| U.S. Department of Energy | DE-FC02-07ER64494 | Nathaniel E Ostrom Christoph Benning |
| European Union Seventh Framework Programme | FP7/2007-2013 n° [627266] | Krzysztof Zienkiewicz |

National Science Foundation     DEB 1737898     Zhi-Yan Du
                                                Natalie Vande Pol
                                                Gregory M Bonito

The funders had no role in study design, data collection and interpretation, or the decision to submit the work for publication.

## Author contributions
Zhi-Yan Du, Conceptualization, Formal analysis, Investigation, Visualization, Methodology, Writing—original draft, Writing—review and editing; Krzysztof Zienkiewicz, Data curation, Formal analysis, Investigation, Visualization, Methodology, Writing—original draft; Natalie Vande Pol, Formal analysis, Investigation, Writing—original draft; Nathaniel E Ostrom, Data curation, Formal analysis, Methodology, Writing—original draft, Writing—review and editing; Christoph Benning, Supervision, Funding acquisition, Methodology, Writing—original draft, Writing—review and editing; Gregory M Bonito, Conceptualization, Formal analysis, Supervision, Funding acquisition, Methodology, Writing—original draft, Project administration, Writing—review and editing

## Author ORCIDs
Zhi-Yan Du ⓘ https://orcid.org/0000-0001-7646-2429
Krzysztof Zienkiewicz ⓘ http://orcid.org/0000-0002-8525-9569
Christoph Benning ⓘ https://orcid.org/0000-0001-8585-3667
Gregory M Bonito ⓘ https://orcid.org/0000-0002-7262-8978

## Decision letter and Author response
Decision letter https://doi.org/10.7554/eLife.47815.030
Author response https://doi.org/10.7554/eLife.47815.031

# Additional files

## Supplementary files
• Transparent reporting form
DOI: https://doi.org/10.7554/eLife.47815.028

## Data availability
All data generated or analyzed during this study are included in the manuscript and supporting files. Source data files have been provided for Supplemental videos. Aligned nucleotide sequences have been submitted to TreeBase (20243).

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
