## [Decision Letter]

Thank you for submitting your work entitled "Algal-fungal symbiosis leads to photosynthetic mycelium" for consideration by *eLife*. Your article has been reviewed by three peer reviewers, one of whom is a member of our Board of Reviewing Editors, and the evaluation has been overseen by a Reviewing Editor and a Senior Editor. The following individuals involved in review of your submission have agreed to reveal their identity: Paola Bonfante (Reviewer #2); Erik Hom (Reviewer #3).

As you will see from the reviews, the reviewers were excited by your discovery of a novel interaction between Nannochloropsis and Mortierella but identified several weaknesses with the experimentation. This led to the consensus decision that the data are not currently strong enough to conclude that these two organisms develop an endosymbiosis. At *eLife*, we invite revision only if additional work required is fairly minor. Unfortunately, this is not the case here. However, we are excited by the topic and if you are able address the criticisms and include additional data to substantiate the claims of endosymbiosis, we would be pleased to consider reviewing a new manuscript (which would be considered a new submission).

*Reviewer #1:*

The authors make an interesting discovery that, when grown together, the alga, Nannochloropsis oceanica and the fungus, Mortierella elongata, appear to develop a consortium. The authors provide some evidence for the movement of carbon and nitrogen between symbionts, some evidence of specificity of the interaction, in that it doesn't occur with just any fungus, and also evidence for the growth of N. oceanica within the M. elongate hyphae, although it is not so clear that the hyphae are still living. The data are intriguing, and the system has potential to provide insight into the initiation of new symbioses and how mutualisms develop, as well as possible applications. However, currently, I consider that the data are not strong enough to support the broad claims that are made and the evidence for endosymbiosis is particularly weak. In several cases, essential controls are missing, and strong claims are made without sufficient experimental support.

1) Interactions of Noc with diverse fungi were assessed on the basis of the appearance of aggregates and as aggregates were observed only for this pair of organisms, these data certainly argue for something unique that occurs during the co-cultivation of these two species. However, quantitative data to show growth of the two organisms over time are lacking, even though it is claimed that they can be co-cultivated together for extended periods and that they benefit from the co-cultivation. There is some evidence that N. oceanica shows a small growth benefit during co-culture in -C or -N medium but not -P. However, the baseline viability percentage is different in these experiments (very high in the -P expt), so it is unclear whether the lack of difference in -P is valid. Additionally, the number of cells that were counted was 5 times less in the P experiment. I was unable to find any data to support the claims that Mortierella 'grows in PBS by incorporating algal-derived metabolites' (claimed in subsection “Nutrient-de1ciency and bene1ts of co-cultivation for *N. oceanica* and *M. elonga”*). In fact, growth data for Mortierella appears to be lacking entirely.

The observations are interesting, but growth data are essential (and preferably in different nutritional conditions) to substantiate the claims of reciprocal nutrient exchange resulting in a mutualism.

2) The idea that the Noc might be growing inside the hyphae is intriguing. The low power light microscope images certainly show instances of algal cells inside the hyphae. However, the data do not convince me that those hyphae are alive. Mainly because in the EM images, the hyphae that clearly contain Noc are empty of organelles. Additionally, the data to support the claim that the Noc is surrounded by a host cell membrane (Figure 4) is very weak. Chemical fixation approaches do not provide the level of resolution needed to support claims of novel membranes. It is very difficult to visualize the membranes in these images and high pressure freezing should be used. Unfortunately, the videos do not provide clear view of Noc inside living hyphae. If they are indeed living together, with Noc inside the Mortierella hyphae, it should be possible to obtain images of hyphae in which there is clear evidence of cytoplasmic streaming along with living algal cells.

The internalization is interesting but needs additional data to substantiate the claims that both of the organisms are indeed alive and growing. If as the authors comment, that cultures can be fragmented and maintained for many months, then it shouldn't be difficult to demonstrate that the mycelium is alive and that the organisms are co-existing.

Additional points

1) As a general comment, some of the experiments are not clearly described. For example, in Figure 1, aggregation and co-occurrence of the algal and fungal cells was observed. It is stated that the interface is reminiscent of lichens but the features on which this conclusion is based are not described. Do Nostoc cells ever aggregate when grown alone in f/2 medium? Are the surface structures on the Nostoc cells present only when grown in co-culture with Mortierella or is this a general feature of Nostoc cells. At a minimum an image of a Nostoc cells grown in the absence of Mortierella should be included. What is the composition of f/2?

2) Figure 2—figure supplement 1. The PCR tests for contamination are a very good idea but positive controls are missing. Do the 3 hyphal samples show a positive PCR reaction with a fungal gene and likewise the same for the Noc?

3) In the labeling experiments, is the radioactivity normalized with respect to dry weight of the organism or with respect to the dry wt of the molecule being measured (eg protein or lipid or carbohydrate). This is important because the proportions of these macromolecules likely differ in the two organisms.

4) Are there differences in the extent of aggregation in nutrient deficient conditions?

5) It is stated that they can be co-cultivations long term. How long?

*Reviewer #2:*

The manuscript by Zhi-Yan Du and colleagues describes a novel interaction established between a biofuel-producing alga (Nannochloropsis oceanica) and the fungus Mortierella elongata. The authors offer a detailed description of the association (starting from the algal aggregation eventually leading to algal internalisation) as well as some functional characterization (including the nutrient exchange between the partners). The authors suggest that their discovery may have relevant biotechnological applications, since both the microbes are important for lipid and fuel production.

The manuscript is sure of interest and novel. It is exciting to see that basal fungi like Mucoromycota may interact not only with most of plant lineages, but also with algae. There are however many weak points in the experimental approaches, in the quality of the electron-microscope pictures as well as in the use of symbiosis-related terminology.

As a general comment (and the authors can or cannot agree) the comparison with lichens is sometimes misleading: the organisation of a lichen is very diverse, with small hyphae which surround algal cells, and very often these hyphae penetrate inside algal cells producing pegs, or intracellular haustoria (see for example Honegger, 1986). The system here described is fully diverse (Noc is very small in diameter, when compared to Mortierella hyphae), and recalls other interactions where algae, as eukaryotic endosymbionts proliferate inside heterotrophic protists like Paramecium, or Hydra. The result of these interactions is a photosynthetic association, where no pseudotissues are produced (differently from lichens). The authors could give a look at Angela Douglas work (2009) or to the extensive review by Nowack and Melkonian, (2010), where these symbioses are illustrated. The main conclusion is that the first function performed by eukaryotic endosymbionts when are involved in stable interactions with living protists is photosynthesis. This general concept could also help the authors to better characterize the functionality of the association they describe (a green-photosynthetic mycelium).

Essential revisions:

- The quality of the pictures revealing the two partner interaction is not fully satisfying. Figure 1 panels C,E,D are very poor. Both hyphae and algal cells seem to be collapsed with material present at the surface which is very difficult to interpret (see for example the surface of the Noc in the panel D). Is this the result of a preparation artefact? What about a control algal cell, which is maintained in the absence of the fungus? is its surface smooth? Are the warts/projections produced in the presence of the fungus? In my opinion a control experiment is missing. N oceanica is usually described with a smooth surface. Also, the legend has to be carefully checked. If the authors say that Noc cells are captured, this implies an active mechanism by the fungus (as for fungi which trap nematodes.). By contrast, the experiments suggest an aggregation (see below).

- Aggregation of algal cells. There are reports demonstrating that N.oceanica can easily aggregate in the presence of bacterial strains or of bio flocculants (Wang et al., 2012). In some cases, the active molecules which act as bioflocculants have been identified (Wan et al., 2012) as proteoglycans.

Since genomics and metabolomics data are available for Mortierella elongata, can the Authors provide some experimental support to the aggregation they describe? Can they provide a time course experiment? The author could treat Nocs with Mortierella exudates just to see whether the aggregation occurs.

- Nutrient transfer experiments are well developed and accurately described. The supplemental material provides many interesting details. However, some aspects are not fully clear. Nutrient experiments: Figure 2 is not easy to interpret. If I well understand Figure 2 A (left), only a reduced quantity of labelled carbon is moving to the fungus (less than 1 radioactivity dpm/mg), while in the text this is described as a relevant quantity. Which is the comparison term to define "relevant" the radioactivity value? Is the difference between the labelled glucose found in attached vs free Noc cells significantly different? In panel C, the same experiment is repeated, but the radioactivity value in the fungus is much higher (12,7%). In addition, why N experiments are represented in a different way?

- Mechanisms underlying the carbon transfer. The results clearly demonstrate that carbon is moving from the alga to the fungus. Does Mortierella genome give some suggestion on the underlying mechanism? Presence of glucose transporter? On the other hand, the results showing a moving of C from the fungus to the alga even in the absence of a physical contact are very difficult to understand. Noc is photosynthetic: have the environmental conditions an impact on its photosynthetic activity? have the Authors checked some photosynthetic parameters under these conditions?

- Nutrient conditions: Noc responds to short and long term N starvation activating specific molecular responses (Dong et al., 2012). Have the Authors checked these recovery mechanisms when Noc cells are maintained in the presence of Mortierella?

- Specificity experiment. This experiment is very interesting, also thinking of the reports which show how bacteria can aggregate the small Noc cells (see previous comment). In my opinion, a couple of information is missing: which is the behaviour of the original Mortierella strain, the strain which contains Mycoavidus? have the Authors checked some lichenised fungi? It would be interesting to see the behaviour of Rhizopus, a related Mucoromycota which hosts B. rhizoxinica. This bacterium enters inside Rhizopus hyphae following modalities which recall those described for Mortierella-Noc, using chitinase to degrade the fungal wall at the tip (Moebius et al., 2014).

In addition, I would comment the negative results on Saccharomyces. Hom and Murray, 2014 clearly demonstrate that the interactions between the yeast and the alga depends on the environmental conditions! the environment (for example, nutrient starvation) is the driving force for the mutualistic association. In this context, the authors should also consider the very interesting results illustrated in Li Chien et al., 2017, where a synthetic platform is developed by associating different yeasts to photosynthetic cyanobacteria.

- Subsection “Long-term co-cultivation leads to internalization of *N. oceanica* within *M. elongata* hyphae”: WGA is a lectin expected to bind to N-acetylglucosamine, the monomer of chitin. Does Noc contain this fungal-wall component?

Looking at the confocal pictures, (Figure 4—Figure supplement 1) no doubt that the WGA is staining the fungal walls, but the algal walls are not labelled (Figure 4—figure supplement 1, panel B), only the division septum between two dividing cells shows a fluorescence. By contrast, the red chlorophyll fluorescence clearly allows to identify both the isolated algal cells and those among the hyphae. In the last panel (on the right) the chloroplast is clearly seen inside the hypha, suggesting the algal internalisation. I would suggest to re-write the description.

Figure 4 has probably to be reorganized in order to allow an easier reading: First the DIC images clearly showing the algal cluster inside at the tip of the hypha, and then a couple of TEM images, selecting the best: I would suggest Figure 4—figure supplement 2 the first two pictures from the left, since the fungus seems to be alive in all the other images, the hyphae are empty: no organelle, no membranes, suggesting that the algae are entering in an empty niche…By contrast in this Figure 4—figure supplement 2 picture, it seems that some fungal membranes are present. And then the magnification of Figure 4 panel C to show the ultrastructure of Noc. However, the pictures do not solve the question whether Noc cells are surrounded by the fungal membrane. Again, the Authors can check their images with algae living inside unicellular protists. For example, the beautiful pictures from Song et al., (2017).

*Reviewer #3:*

This manuscript documents a very exciting finding: the endosymbiosis of Nannochloropsis algal cells by a Mortierella fungus in the Mucoromycota, a phylum that is coming under greater scrutiny in relation to the evolution of plant-fungal associations. The text is well written, and the experiments described are compelling and demonstrate the intracellular association of the alga (via light and electron microcopy) and the exchange of carbon and nitrogen between the alga and fungus. While the mechanisms underlying the (vertical?) transmission and maintenance of this new co-culture 'induced' symbiosis are not explored, this work establishes the basis for further research.

A few concerns/questions:

1) In the Abstract, it is written: "This symbiosis begins with chemotactic attraction…". It's not clear to me what evidence there is for this claim. Is this statement based on the aggregation of algal cells at the tips of fungal hyphae? Can one safely conclude that there was chemotactic attraction based on this?

2) Is there any series of images or a movie that be provided that shows how the dense clustering of the algal cells at hyphal tips changes or progresses over time (Discussion section)? Perhaps from lower density to higher density?

3) Video 4 is referenced (subsection “Long-term co-cultivation leads to internalization of *N. oceanica* within *M. elongata* hyphae”) as providing evidence that both algal and fungal cells can be passaged through fragmentation and remain viable. This video shows a focal hypha with endosymbiotic algae (within the context of a larger "green tissue") time-lapsed over 6 days (not months of co-culture) but does not show any fragmentation and/or passaging. Was the wrong file uploaded? Is there another video or set of figures/images that can actually be used to support the stated claim?

[Editors’ note: what now follows is the decision letter after the authors submitted for further consideration.]

Thank you for submitting your article "Algal-fungal symbiosis leads to photosynthetic mycelium" for consideration by *eLife*. Your article has been reviewed by three peer reviewers, one of whom is a member of our Board of Reviewing Editors, and the evaluation has been overseen by a Reviewing Editor and Ian Baldwin as the Senior Editor. The following individuals involved in review of your submission have agreed to reveal their identity: Maria J Harrison (Reviewer #1); Paola Bonfante (Reviewer #2); Erik Hom (Reviewer #3).

The reviewers have discussed the reviews with one another and the Reviewing Editor has drafted this decision to help you prepare a revised submission.

Summary:

The authors have made an interesting discovery that the alga, *Nannochloropsis oceanica* and the fungus, *Mortierella elongata*, are capable of living together in a mutualistic association which includes growth of Nannochloropsis within the Mortierella hyphae. This is a unique study has potential to provide insights into the emergence of mutualisms and possibly endosymbiosis. All three reviewers appreciate the authors' efforts to add new data and agree that the main claims are now well supported. The revisions outlined below are essential but should not be too onerous as they are largely text edits to ensure that all claims are fully supported by the data presented and that the assumptions being made are clearly stated.

Essential revisions:

Figure 4G. The authors have tempered their comments about the fungal membranes, and we appreciate this. However, the TEM Pictures (Figure 4) do not fully solve the question: are the algal cell surrounded by a fungal-derived membrane? We feel that this cannot be concluded from the data presented. In many cases, it is apparent that the Mortierella hyphae are empty and the fact that the fungus grows when put in a new plate simply means that some hyphae are alive and capable of re-starting their growth. But these are not necessarily the hyphae with Noc in them. So, unless you have data to show this, we request that you delete the sentence that says that 'putative fungal membranes surround internalized algal cells.'

We really appreciate the new scanning images showing the Nannochloropsis cell wall structure. And indeed, they reveal that the alga alone has a smooth surface, as reported in literature. However, in order to conclude that the extensions exist underneath the smooth outer coat, transmission electron microscopy of cross sections are needed. It is possible that the extensions have been elicited by the contact with the fungus. Consequently, we request that the conclusions be modified and that this point is discussed (unless you can include cross sections that show the extensions below the coat).

Abstract: The first sentence does not make sense (…the coevolution of land plants and lichens). The abstract has been corrected following previous suggestion in some documents, but not in the PDF "merged new version". In addition, I believe that the term coevolution requires two members. (for example coevolution between plants and fungi, between fungi and insects.). In conclusion: the sentence has to be re-written.

Discussion section: hyphal tips are among the least developed tissues. Please note that tissue means: groups of cells that have a similar structure and origin and act together to perform a specific function. Therefore, the term tissue cannot be use for a part of a cell (hypha in this case)..In this context I would write: the hyphal tips are the least differentiated portions of a mycelial network

Subsection “Nutrient-de1ciency and bene1ts of co-cultivation for *N. oceanica* and *M. elongate”*: Somewhere here, we think it's proper to acknowledge that the linolenic acid marker for biomass was standardized under replete, monoculture conditions. The authors' have not ruled out the possibility that marker correspondence with biomass may break down under the conditions of co-culture, especially since lipid compositional remodeling is certainly possible and not unprecedented in symbioses (e.g., between plants and arbuscular mycorhizal fungi). We think it right to add a simple, honest sentence that states that insignificant changes in C18:3 vs.biomass are assumed for any physiological changes that might be experienced by M. elongata in coculture. (I think it probably is insignificant, but one should be precise and not speculate.) This applies to Figure 3—figure supplement 2 as well.

The nutrient exchange experiments are nicely performed and presented, but a discussion of their biological meaning is missing. M.elongata is a strong saprotroph and does not need the carbon coming from the alga. On the other hand, it also releases N to the alga. So, what is the benefit for the fungus? The only benefit seems to be the increased fungal biomass; however, this is contingent on the point noted above. The C18:3 quantification could mirror a different metabolism (lipid store) more than an active growth. Please add a point of discussion including the C and N conditions of the media in which these experiments were performed.

Subsection “Long-term co-cultivation leads to internalization of *N. oceanica* within *M. elongata* hyphae”: for algae that are inside live fungi, how can you be sure they are really alive based only on these data? Free/extracellular algae might be accessible to SYTOX Green, but algae in live fungi may never be exposed to SYTOX Green, yes? (The live fungi would exclude it, so one may not be able to tell intracellular viability of the algae.) We do not feel this is a major point because the EM data indicate that algae are alive, but it might not be possible to determine true viability of intracellular algae with exclusionary dyes and this should be acknowledged.

Abstract – Long term co-cultivation is a subjective term. The length of time observed should be inserted here.

Figure 3—figure supplement 2. Again, details of the statistical tests used should be added.

Figure 4 G -What is Mo? It is missing from the legend.

Subsection “Long-term co-cultivation leads to internalization of *N. oceanica* within *M. elongata* hyphae”: We suggest rewording to be more conservative in claims: "The results is consistent with the notion that both fungal host and algae inside are alive (Figure 4—figure supplement 5), although DIC microscopy…surrounded by fungal organelles and what appear to be lipid droplets…"

Changed "is consistent with" for "showed", added "although" before "DIC microscopy, removed "the" before "fungal organelles", and changed "and what appear to be" for "such as". Lipid droplets are not fungal organelles…. As a side note: how does one know that these are lipid droplets? Do they stain with Nile Red? This is stated but not justified.

Discussion section: We suggest rewording to be more conservative in claim: "green hyphae appear to remain alive after 2-months…". Changed "appear to remain" for "remained".

Please insert the nature of the statistical tests used in the legends and/or in the Materials and methods section.

---

## [Author Response]

We first want to thank you for your constructive reviews of our manuscript and say that we found your comments valuable. Following your advice, we carried out many additional experiments that were suggested, including a series of new scanning electron microscopy experiments to detail the cell wall interaction between the two cell types, PCR controls for isolation of algae from fungi, photosynthesis/chlorophyll assays of cocultured algae, growth/biomass assays of co-cultured algae and fungi, light and confocal microscopy and viability assays of both algae and fungi in green hyphae. We have prepared new figures and revised all of the figures for clarity, including new transmission electron micrographs showing the endo-microalgae surrounded by fungal membranes and vesicles. We feel your suggestions and these additions have greatly improved our new submission and hope that you and the reviewers concur.

Reviewer #1:

The authors make an interesting discovery that, when grown together, the alga, Nannochloropsis oceanica and the fungus, Mortierella elongata, appear to develop a consortium. The authors provide some evidence for the movement of carbon and nitrogen between symbionts, some evidence of specificity of the interaction, in that it doesn't occur with just any fungus, and also evidence for the growth of N. oceanica within the M. elongate hyphae, although it is not so clear that the hyphae are still living. The data are intriguing, and the system has potential to provide insight into the initiation of new symbioses and how mutualisms develop, as well as possible applications. However, currently, I consider that the data are not strong enough to support the broad claims that are made and the evidence for endosymbiosis is particularly weak. In several cases, essential controls are missing, and strong claims are made without sufficient experimental support.

We have carried out additional experiments to support the findings and we have toned down strong language.

1) Interactions of Noc with diverse fungi were assessed on the basis of the appearance of aggregates and as aggregates were observed only for this pair of organisms, these data certainly argue for something unique that occurs during the co-cultivation of these two species. However, quantitative data to show growth of the two organisms over time are lacking, even though it is claimed that they can be co-cultivated together for extended periods and that they benefit from the co-cultivation. There is some evidence that N. oceanica shows a small growth benefit during co-culture in -C or -N medium but not -P. However, the baseline viability percentage is different in these experiments (very high in the -P expt), so it is unclear whether the lack of difference in -P is valid. Additionally, the number of cells that were counted was 5 times less in the P experiment. I was unable to find any data to support the claims that Mortierella 'grows in PBS by incorporating algal-derived metabolites' (claimed in subsection “Nutrient-de1ciency and bene1ts of co-cultivation for N. oceanica and M. elonga”). In fact, growth data for Mortierella appears to be lacking entirely.The observations are interesting, but growth data are essential (and preferably in different nutritional conditions) to substantiate the claims of reciprocal nutrient exchange resulting in a mutualism.

Because P was not shown to be important in the interaction and is not a major part of this research we have removed the result from the manuscript and prepared a new Figure 3 and new Figure 3—figure supplement 1.

Quantification of fungal biomass is technically challenging. We did perform time-lapse microscopy and we found that the fungal hyphae kept growing in PBS buffer in the presence of the algae partner, whereas the fungi alone in PBS didn’t show any obvious new growth. We provide a side-by-side video to show the difference (Video 1).

In this new submission we have made great efforts to quantify the growth of the algae and fungi during co-cultivation. We have found that *Mortierella elongata* has major unique fatty acids such as the α-linolenic acid (C18:3) that are only a minor component in *Nannochloropsis*. Therefore, it can serve as a biomarker to assess fungal abundance. We onserved that the ratio of C18:3 to biomass in the *Mortierella* fungus is very consistent following the incubation in f/2-N medium (Figure 3—figure supplement 2A). Thus, we used C18:3 as a marker to quantify the fungal biomass within algae-fungi aggregates. We measured the C18:3 content and total biomass of algae-fungi aggregates. We did observe increase in the biomass of both algae and fungi following co-cultivation in f/2-N medium (nutrient-deprived medium for both fungi and algae), which was not observed when partners were incubated alone in the same medium (Figure 3—figure supplement 2B).

2) The idea that the Noc might be growing inside the hyphae is intriguing. The low power light microscope images certainly show instances of algal cells inside the hyphae. However, the data do not convince me that those hyphae are alive. Mainly because in the EM images, the hyphae that clearly contain Noc are empty of organelles. Additionally, the data to support the claim that the Noc is surrounded by a host cell membrane (Figure 4) is very weak. Chemical fixation approaches do not provide the level of resolution needed to support claims of novel membranes. It is very difficult to visualize the membranes in these images and high pressure freezing should be used. Unfortunately, the videos do not provide clear view of Noc inside living hyphae. If they are indeed living together, with Noc inside the Mortierella hyphae, it should be possible to obtain images of hyphae in which there is clear evidence of cytoplasmic streaming along with living algal cells.The internalization is interesting but needs additional data to substantiate the claims that both of the organisms are indeed alive and growing. If as the authors comment, that cultures can be fragmented and maintained for many months, then it shouldn't be difficult to demonstrate that the mycelium is alive and that the organisms are co-existing.

We understand the reviewers’ concerns. To check whether the algae cells are living inside the fungal hyphae and whether the fungal hypha are themselves living (have cellular contents), we have performed additional microscopy including DIC light, confocal, and electron microscopy. Our new data demonstrate that algal cells are alive within living fungal cells.

In this revision, we have replaced the TEM images in Figure 4 with one showing that intact algal cells are inside the fungal hypha, and they are surrounded by fungal cytoplasm including organelles (AG77 and Noc, Figure 4F and G; NVP64 and Noc, Figure 4—figure supplement 2C).

Confocal SYTOX green microscopy images of Figure 4—figure supplement 5 A-C demonstrate that both internalized algae and host cells are alive. The SYTOX probe enters dead cells highlighting them and the signal is obvious when dead cells are used as controls (Figure 4—figure supplement 5D to H). Several dead algae cells were observed in the aggregates, but none were observed inside the fungal hyphae. We then used DIC light microscope to check whether the fungal hyphae were hollow, or whether they had cytoplasm and organelles. We have observed that the fungal hyphae contain cytoplasm and the algal cells within fungal hyphae are surrounded by fungal vesicles, including lipid droplets, which are present in living cells (Figure 4—figure supplement 6). We recorded real-time videos to show the 3D content of the green hyphae and Video 5 is a representative one. We also note that cytoplasmic streaming is common in young actively growing cells in low carbon f/2 medium, but after 2-3 weeks cytoplasmic streaming is not evident in older parts of the mycelial network. Hence, it is not unexpected that no cytoplasmic streaming was observed in the mature fungal hyphae over one-month of co-cultivation.

Essential revisions:1) As a general comment, some of the experiments are not clearly described. For example, in Figure 1, aggregation and co-occurrence of the algal and fungal cells was observed. It is stated that the interface is reminiscent of lichens but the features on which this conclusion is based are not described. Do Nostoc cells ever aggregate when grown alone in f/2 medium? Are the surface structures on the Nostoc cells present only when grown in co-culture with Mortierella or is this a general feature of Nostoc cells. At a minimum an image of a Nostoc cells grown in the absence of Mortierella should be included. What is the composition of f/2?

We thank the reviewer for these comments. We have revised the manuscript accordingly to provide more detailed descriptions. To clarify, we are working with *Nannochloropsis* (Noc), a unicellular microalga rather than the cyanobacterium *Nostoc*. We demonstrate through additional controlled experiments that the morphology and aggregation of *Nannochloropsis* (Noc) cells differ significantly between mono-culture and co-culture with living *Mortierella.* Following prolonged incubation after stationary phase in f/2 medium *Nannochloropsis* (Noc) cells can aggregate when grown alone, but in comparatively small clusters (up to dozens of cells) (Figure 1—figure supplement 1A). Through scanning electron microscopy (SEM) we demonstrate that Noc cells incubated as a monoculture have a smooth outer cell wall surface (Figure 1D; Figure 1-S1A), whereas Noc cells co-cultured with living *Mortierella* cells exhibit a fibrous extension from their cell walls (Figure 1C, E and F; Figure 1—figure supplement 1D). Specifically, the thin outer layer breaks away from Noc cells when incubated with *Mortierella*, and the fibrous extensions beneath become exposed (Figure 1—figure supplements1D). Large pieces of the broken membrane are evident in Noc cells (yellow arrows, Figure 1—figure supplement 1D), while the others have smaller residual fragments (Figure 1E and F). We also examined the free Noc cells in the supernatant of Noc-*Mortierella* co-culture, and we found the cells have a partially damaged cover membrane (not as smooth as in the control) but the fibrous extensions were not exposed (Figure 1—figure supplement 1C).

As in control algal mono-cultures, the thin outer membrane covering Noc cells is also evident when co-cultured with non-interactive fungi (*Clavulina* sp. PMI390 and *Morchella americana* GB760 – Figure 3—figure supplement 3 and Figure 3—figure supplement 4) or with dead *Mortierella* cells (Figure 1—figure supplement 1B). The fibrous extensions seem to be important in the physical interaction of Noc and AG77 cells. We have included these new findings in the manuscript. We have also included the composition of f/2 medium in the method section, which is a common and widely used general enriched medium for growing marine algae.

2) Figure 2—figure supplement 1. The PCR tests for contamination are a very good idea but positive controls are missing. Do the 3 hyphal samples show a positive PCR reaction with a fungal gene and likewise the same for the Noc?

We have performed PCR tests with positive and negative PCR controls. The three hyphal samples showed a positive PCR reaction for the fungal marker gene *EF1a*, and the two algal samples had a positive amplification of the algal gene *AURE04*. There was no PCR amplification for either negative control reactions (Figure 2—figure supplement 1D and E).

3) In the labeling experiments, is the radioactivity normalized with respect to dry weight of the organism or with respect to the dry wt of the molecule being measured (eg protein or lipid or carbohydrate). This is important because the proportions of these macromolecules likely differ in the two organisms.

For the labelling experiments we used total dry weight of the cells. We now have clear description of this in the Method section and clarify this in the figure legend.

4) Are there differences in the extent of aggregation in nutrient deficient conditions?

We did not see any obvious difference in the Noc-AG77 aggregates following nutrient deficient conditions.

5) It is stated that they can be co-cultivations long term. How long?

We define long-term co-culture as lasting between one to three months. Both Noc and AG77 are fast growers. Noc cells can double within 48 hours and AG77 can fill a plate or extend across a flask within 2-3 days after inoculation.

Reviewer #2:

The manuscript by Zhi-Yan Du and colleagues describes a novel interaction established between a biofuel-producing alga (Nannochloropsis oceanica) and the fungus Mortierella elongata. The authors offer a detailed description of the association (starting from the algal aggregation eventually leading to algal internalisation) as well as some functional characterization (including the nutrient exchange between the partners). The authors suggest that their discovery may have relevant biotechnological applications, since both the microbes are important for lipid and fuel production.1) The manuscript is sure of interest and novel. It is exciting to see that basal fungi like Mucoromycota may interact not only with most of plant lineages, but also with algae. There are however many weak points in the experimental approaches, in the quality of the electron-microscope pictures as well as in the use of symbiosis-related terminology.

We have improved our electron-micrographs, and reassessed terminology following the reviewer’s comments.

2) As a general comment (and the Authors can or cannot agree) the comparison with lichens is sometimes misleading: the organisation of a lichen is very diverse, with small hyphae which surround algal cells, and very often these hyphae penetrate inside algal cells producing pegs, or intracellular haustoria (see for example Honegger, 1986). The system here described is fully diverse (Noc is very small in diameter, when compared to Mortierella hyphae), and recalls other interactions where algae, as eukaryotic endosymbionts proliferate inside heterotrophic protists like Paramecium, or Hydra. The result of these interactions is a photosynthetic association, where no pseudotissues are produced (differently from lichens). The authors could give a look at Angela Douglas work (2009) or to the extensive review by Nowack and Melkonian, (2010), where these symbioses are illustrated. The main conclusion is that the first function performed by eukaryotic endosymbionts when are involved in stable interactions with living protists is photosynthesis. This general concept could also help the authors to better characterize the functionality of the association they describe (a green-photosynthetic mycelium).

We thank the reviewer for these suggestions and agree. We have revised and included these in the Discussion section.

Major comments:3) The quality of the pictures revealing the two partner interaction is not fully satisfying. Figure 1 panels C,E,D are very poor. Both hyphae and algal cells seem to be collapsed with material present at the surface which is very difficult to interpret (see for example the surface of the Noc in the panel D). Is this the result of a preparation artefact? What about a control algal cell, which is maintained in the absence of the fungus? is its surface smooth? Are the warts/projections produced in the presence of the fungus? In my opinion a control experiment is missing. N oceanica is usually described with a smooth surface. Also, the legend has to be carefully checked. If the authors say that Noc cells are captured, this implies an active mechanism by the fungus (as for fungi which trap nematodes.). By contrast, the experiments suggest an aggregation (see below).

We thank the reviewer for these suggestions.

Following these suggestions, we have carried out controlled experiments to assay algal cells grown in mono-culture in f/2 medium, and also with non-living *Mortierella* and non-compatible fungi. As mentioned above, we found that these Noc cells alone have a smooth surface (Figure 1D; Figure 1—figure supplement 1A) when grown in mono-culture or with non-living *Mortierella* or non-compatible fungi (e.g. *Morchella, Clavulina* spp.). Fibrous extensions were only observed in the Noc cells co-cultured with live *Mortierella* cells (Figure 1C, E and F; Figure 1—figure supplement 1D). The fibrous extensions are covered by a thin layer of relatively smooth membrane. When incubated with *Mortierella* this layer breaks up exposing the fibrous extensions below (Figure 1—figure supplement 1D). Some Noc cells still have big pieces of the broken membrane (yellow arrows, Figure 1—figure supplement 1D), while the others have small residues (Figure 1E and F).

Scholz et al., 2014 performed super high resolution Cryo-EM and they suggested that the fibrous extensions were on the outer layer of *Nannochloropsis gaditana*, a related algal species. However, according to our data, it appears the fibrous extensions are exposed after the cover membrane is broken. In fact, the cover membrane is also visible in the Cryo-EM images by Scholz et al., 2014. The authors may have missed the membrane as a part of the Noc cell wall because of how they processed and prepared their samples (Figure 2B and C in Scholz et al., 2014). Further research on *N. gaditana* also showed that the cells have a smooth surface by SEM (Jazzar et al., 2015).

We agree with the reviewer’s comments on the legend and have also changed the terminology as an ‘aggregation’ rather than captured cells.

4) Aggregation of algal cells. There are reports demonstrating that N.oceanica can easily aggregate in the presence of bacterial strains or of bio flocculants (Wang et al., 2012). In some cases, the active molecules which act as bioflocculants have been identified (Wan et al., 2012) as proteoglycans.Since genomics and metabolomics data are available for Mortierella elongata, can the Authors provide some experimental support to the aggregation they describe? Can they provide a time course experiment? The author could treat Nocs with Mortierella exudates just to see whether the aggregation occurs.

We thank the reviewer for these suggestions. We think the important findings in this manuscript are the details of the physical cell wall interaction and mutualism between the algae and fungi, as well as long-term co-culture that led to the formation of green hyphae. In terms of the algae flocculation by fungi, we have recently published on the ability of *Mortierella* to bioflocculate *Nannochloropsis* as an efficient, cost effective, and environment friendly approach to harvest microalgae compared to conventional harvesting methods such as centrifuge and chemical flocculation. We provide data including a short time course flocculation and lipid/fatty acid profiling results, as well as other oil productivity data for biotechnological purposes (Du et al., 2018). No aspect of nutrient exchange or green hyphae observed after longer term incubations were discussed in the *Biotechnology for Biofuels* paper, as these data are the substance of this submission. Determining the composition of the algal outer wall, and the means by which *Mortierella* affects this, is part of a larger ongoing JGI supported project, and beyond the scope of the current study.

5) Nutrient transfer experiments are well developed and accurately described. The supplemental material provides many interesting details. However, some aspects are not fully clear. Nutrient experiments: Figure 2 is not easy to interpret. If I well understand Figure 2 A (left), only a reduced quantity of labelled carbon is moving to the fungus (less than 1 radioactivity dpm/mg), while in the text this is described as a relevant quantity. Which is the comparison term to define "relevant" the radioactivity value? Is the difference between the labelled glucose found in attached vs free Noc cells significantly different? In panel C, the same experiment is repeated, but the radioactivity value in the fungus is much higher (12,7%). In addition, why N experiments are represented in a different way?

First of all, we thank you for your comment and have tried to more clearly describe the presented data and figures. One key point is that the majority of transferred ^14^C-carbon ended up in the lipid fraction of the fungal cells. The later relative values in Figure 2 panels C and D, are the total transferred ^14^C-carbon in different kinds of samples such as non-labelled fungi with physical contact to labelled algae (Figure 2 panel C #1, 12.7%) and with physical contact to heat-killed-labelled algae (Figure 2 panel C #3, 1.3%) compared to the total radioactivity of prelabelled samples such as the ^14^C-labelled algae (Figure 2 panel C, 100%). These analyses were aimed to test whether physical contact and live cells are essential to the ^14^C-carbon transfer. We have revised the text content and the figure to make it easier to understand. We add “100%” after the ^14^C-Noc and ^14^C-AG77 in Figure 2 panels C and D.

Compared to the carbon exchange experiments that use ^14^C-carbon that can be measured with a scintillation counter and readily presented, the nitrogen exchange experiment was conducted using ^15^N, a stable, non-radioactive isotope of nitrogen in combination isotope ratio mass spectrometry to calculate [^15^N/(^15^N+ ^14^N), mol/mol]. The latter experiments are admittedly more complicated than the carbon labelling results (Figure 2—figure supplement 3J and K). To make these results better understandable to the general audience, we summarize the results in a simple figure (Figure 2 panel E), and provide more details in the supplementary files.

6) Mechanisms underlying the carbon transfer. The results clearly demonstrate that carbon is moving from the alga to the fungus. Does Mortierella genome give some suggestion on the underlying mechanism? Presence of glucose transporter? On the other hand, the results showing a moving of C from the fungus to the alga even in the absence of a physical contact are very difficult to understand. Noc is photosynthetic: have the environmental conditions an impact on its photosynthetic activity? have the Authors checked some photosynthetic parameters under these conditions?

We agree these are interesting and obvious remaining questions, but decided that to address the mechanism of carbon transport should be the next step requiring additional metabolic, transcriptomic experiments, and a transformation system for the fungus in place. We believe these experiments go beyond the current manuscript, which already contains a large amount of original data.

Regarding C transfer from fungus to algae we report that *Mortierella* fungi release organic carbon and nitrogen to the environment/medium when they are incubated alone in the f/2 medium (Figure 3D and E). This would allow the unattached algae to receive carbon and nitrogen from the co-cultured fungi (Figure 2).

Environmental conditions are sure to impact photosynthesis. We analyzed the chlorophyll content that can be extracted using acetone:DMSO solvents, as this can be used as a proxy of photosynthetic potential. The chlorophyll content decreases when Noc cells are stressed. We did see a significantly higher chlorophyll content in the Noc cells co-cultured with AG77 fungi than in the Noc cells alone serving as control after 10-days-prolonged incubation in f/2 medium, indicating that the Noc cells with *Mortierella* fungi have higher photosynthetic potential (Figure 3—figure supplement 1A).

7) Nutrient conditions: Noc responds to short and long term N starvation activating specific molecular responses (Dong et al., 2012). Have the Authors checked these recovery mechanisms when Noc cells are maintained in the presence of Mortierella?

We observed increased viability of Noc cells following long term (10 days) N starvation in the presence of *Mortierella* fungi (Figure 3A to C), which release organic carbon and nitrogen to the medium (Figure 3D and E). The Noc cells without *Mortierella* were significantly more stressed, evident by chlorophyll degradation following prolonged incubation (Figure 3—figure supplement 1A).

8) Specificity experiment. This experiment is very interesting, also thinking of the reports which show how bacteria can aggregate the small Noc cells (see previous comment). In my opinion, a couple of information is missing: which is the behaviour of the original Mortierella strain, the strain which contains Mycoavidus? have the Authors checked some lichenised fungi? It would be interesting to see the behaviour of Rhizopus, a related Mucoromycota which hosts B. rhizoxinica. This bacterium enters inside Rhizopus hyphae following modalities which recall those described for Mortierella-Noc, using chitinase to degrade the fungal wall at the tip (Moebius et al., 2014).In addition, I would comment the negative results on Saccharomyces. Hom and Murray, 2014 clearly demonstrate that the interactions between the yeast and the alga depends on the environmental conditions! the environment (for example, nutrient starvation) is the driving force for the mutualistic association. In this context, the authors should also consider the very interesting results illustrated in Li Chien et al., 2017, where a synthetic platform is developed by associating different yeasts to photosynthetic cyanobacteria.

We thank the reviewer for these suggestions. We have tested the wild-type *Mortierella elongata* AG77, which carries the endobacterium *Mycoavidus cysteinexigens*. We observed significantly reduced viability of Noc cells with the WT AG77 compared to the cured AG77, as well as severe chlorophyll degradation and ROS accumulation in the Noc cells when co-cultured with the WT AG77. However, we have repeated the experiments several times and the results were not consistent. This could result from fluctuations in bacterial activity. While intriguing, the presence of the endobacteria makes the symbiosis more complicated and is not the focus of this research. Thus, we removed the WT AG77 data and we have revised the Discussion section. Although we did not screen specifically for *Rhizopus* or lichen fungi, we did screen a phylo-diverse selection of fungi including relatives of these organisms. We found the phenotype we report here to be unique to *Mortierella*.

9) Subsection “Long-term co-cultivation leads to internalization of N. oceanica within M. elongata hyphae”: WGA is a lectin expected to bind to N-acetylglucosamine, the monomer of chitin. Does Noc contain this fungal-wall component?

Yes. It has been reported that Noc has the N-acetylglucosamine in the cell wall (Scholz et al., 2014), and we do know that *Mortierella* is efficient at degrading this substrate.

10) Looking at the confocal pictures, (Figure 4—Figure supplement 1) no doubt that the WGA is staining the fungal walls, but the algal walls are not labelled (Figure 4—figure supplement 1, panel B), only the division septum between two dividing cells shows a fluorescence. By contrast, the red chlorophyll fluorescence clearly allows to identify both the isolated algal cells and those among the hyphae. In the last panel (on the right) the chloroplast is clearly seen inside the hypha, suggesting the algal internalisation. I would suggest to re-write the description.

We thank the reviewer for these suggestions. The Figure 4—figure supplement 1B is not just a regular confocal image focused on a single panel. It’s a stacked confocal image which shows the 3D-view of the WGA and chlorophyll signals. That’s the reason why the WGA signal is all over the cells instead of just a circle. The division septum did show a stronger signal. We lowered the gain of WGA to show the inside chlorophyll signal. Otherwise, they will be bright green cells. We have a video of a rotating 3D-model to show how the confocal images are stacked, as well as for the Figure 4—figure supplement 1C (Video 2). We have rewritten the description as requested.

11) Figure 4 has probably to be reorganized in order to allow an easier reading: First the DIC images clearly showing the algal cluster inside at the tip of the hypha, and then a couple of TEM images, selecting the best: I would suggest Figure 4—figure supplement 2 the first two pictures from the left, since the fungus seems to be alive..in all the other images, the hyphae are empty: no organelle, no membranes, suggesting that the algae are entering in an empty niche…By contrast in this Figure 4—figure supplement 2 picture, it seems that some fungal membranes are present. And then the magnification of Figure 4 panel C to show the ultrastructure of Noc. However, the pictures do not solve the question whether Noc cells are surrounded by the fungal membrane. Again, the authors can check their images with algae living inside unicellular protists. For example, the beautiful pictures from Song et al., (2017).

We thank the reviewer for the suggestions. We first replaced the TEM images in Figure 4 with the ones showing that intact algal cells are inside the fungal hypha and they are surrounded by fungal membranes and organelles (AG77 and Noc, Figure 4F and G; NVP64 and Noc, Figure 4—figure supplement 2C). The previous TEM images showing algal cells inside fungal hypha with clear ultrastructure of chloroplast and some other algal organelles were moved to the Figure 4—figure supplement 2B. The algal cells have unique and distinguishable chloroplast and thylakoid membranes. For the fungal organelles, we include a TEM control of the fungi incubated alone in f/2 medium to compare the algal and fungal organelles (Figure 4—figure supplement 2A). Some fungal mycelia are hollow after the harsh steps of TEM sample preparation as the other images of co-cultured fungal tissue.

Unfortunately, we were unsuccessful with Cyro-EM to get higher resolution images to address whether a fungal membrane surrounds the algal cells. However, we have used DIC light and confocal microscopy with SYTOX green and demonstrated algae cells are living inside living fungal cells. Figure 4—figure supplement 5A to C is a representative example that both the algae and hyphae are alive. The SYTOX probe enters only the dead cells and the signal is obvious in the dead cell controls (Figure 4—figure supplement 5D to H).

We then used DIC light microscopy and have observed that the algal cells inside fungal hyphae are surrounded by fungal vesicles and lipid droplets, which are usually presented in live cells (Figure 4—figure supplement 6). Given that the samples are usually not in the same focal panel at high magnification, we provide videos to show the 3D content of the green hyphae and Video 5 is a representative one.

Reviewer #3:

This manuscript documents a very exciting finding: the endosymbiosis of Nannochloropsis algal cells by a Mortierella fungus in the Mucoromycota, a phylum that is coming under greater scrutiny in relation to the evolution of plant-fungal associations. The text is well written, and the experiments described are compelling and demonstrate the intracellular association of the alga (via light and electron microcopy) and the exchange of carbon and nitrogen between the alga and fungus. While the mechanisms underlying the (vertical?) transmission and maintenance of this new co-culture 'induced' symbiosis are not explored, this work establishes the basis for further research.Essential revisions:1) In the Abstract, it is written: "This symbiosis begins with chemotactic attraction…". It's not clear to me what evidence there is for this claim. Is this statement based on the aggregation of algal cells at the tips of fungal hyphae? Can one safely conclude that there was chemotactic attraction based on this?

We have rephrased this. The Nannochloropsis-Mortierella interaction begins with the flocculation of algal cells with the mycelium and the loss of the outer cell wall covering in the algal photobiont.

2) Is there any series of images or a movie that be provided that shows how the dense clustering of the algal cells at hyphal tips changes or progresses over time (Discussion section)? Perhaps from lower density to higher density?

This is a great point. We have spent tremendous efforts to try to address this, but it is a formidable challenge. We can observe the green hyphae in the long-term co-culture and green hyphae are distinguishable under DIC microscope (Figure 4B to E, Figure 4—figure supplement 4). However, to record the progress through time-lapse is extremely difficult. This is due to the fact that the process occurs over weeks, and the focal plane is quite restricted with light microscopy. Given it is a living growing system (3D), we are not able to predict where in the colony these hyphae will develop, and what point of the 3D colony to focus on over days. For instance, Figures 4-S3C to F were taken as time-lapse images over 3 days at a randomly selected region. It took us thousands of hours of microscopy time to observe this. To complicate matters, the algae and fungi form aggregates after co-culture (Figure 4—figure supplement 1D and E), and if the cells are too dense in number, visibility is obstructed. Nonetheless, we were very lucky to record a series of images showing a group of algal cells attached to the hyphal tip growing inside the hyphae (Figure 4—figure supplement 3A and B).

3) Video 4 is referenced (subsection “Long-term co-cultivation leads to internalization of N. oceanica within M. elongata hyphae”) as providing evidence that both algal and fungal cells can be passaged through fragmentation and remain viable. This video shows a focal hypha with endosymbiotic algae (within the context of a larger "green tissue") time-lapsed over 6 days (not months of co-culture) but does not show any fragmentation and/or passaging. Was the wrong file uploaded? Is there another video or set of figures/images that can actually be used to support the stated claim?

The reviewer is correct. Video 4 shows the growth and dividing of algal cells within the fungal hyphae, not fragmentation. However, the co-culture in the video was fragmented (vegetative propagated) from an older mother colony, as is commonly done with fungal mycelium in the lab. We have revised the manuscript to read: “While there is no indication that algae are transmitted vertically through fungal reproductive structures, the algal cells remain viable (growing and dividing) over months of co-culture (Video 4).”

[Editors’ note: The responses to the re-review follow.]

Essential revisions:Figure 4G. The authors have tempered their comments about the fungal membranes, and we appreciate this. However, the TEM Pictures (Figure 4) do not fully solve the question: are the algal cell surrounded by a fungal-derived membrane? We feel that this cannot be concluded from the data presented. In many cases, it is apparent that the Mortierella hyphae are empty and the fact that the fungus grows when put in a new plate simply means that some hyphae are alive and capable of re-starting their growth. But these are not necessarily the hyphae with Noc in them. So, unless you have data to show this, we request that you delete the sentence that says that 'putative fungal membranes surround internalized algal cells.'

We have deleted the sentence.

We really appreciate the new scanning images showing the Nannochloropsis cell wall structure. And indeed, they reveal that the alga alone has a smooth surface, as reported in literature. However, in order to conclude that the extensions exist underneath the smooth outer coat, transmission electron microscopy of cross sections are needed. It is possible that the extensions have been elicited by the contact with the fungus. Consequently, we request that the conclusions be modified and that this point is discussed (unless you can include cross sections that show the extensions below the coat).

We added an explanation to the Results section as follows: “While it is possible that the fibrous extensions have been elicited by the contact with the fungus, remnant pieces of the outer coat covering the underlying extensions are evident in our observations, as shown in Figure 1E and F and Figure 1—figure supplement 1D. Therefore, it seems likely that these extensions are present underneath the outer smooth layer.”

Abstract: The first sentence does not make sense (…the coevolution of land plants and lichens). The abstract has been corrected following previous suggestion in some documents, but not in the PDF "merged new version". In addition, I believe that the term coevolution requires two members. (for example, coevolution between plants and fungi, between fungi and insects.). In conclusion: the sentence has to be re-written.

We agree with the reviewers and have changed ‘coevolution’ to ‘evolution’.

Discussion section: hyphal tips are among the least developed tissues. Please note that tissue means: groups of cells that have a similar structure and origin and act together to perform a specific function. Therefore, the term tissue cannot be use for a part of a cell (hypha in this case). In this context I would write: the hyphal tips are the least differentiated portions of a mycelial network

We agree, and the sentence has been re-written as suggested.

Subsection “Nutrient-de1ciency and bene1ts of co-cultivation for N. oceanica and M. elongate”: Somewhere here, we think it's proper to acknowledge that the linolenic acid marker for biomass was standardized under replete, monoculture conditions. The authors' have not ruled out the possibility that marker correspondence with biomass may break down under the conditions of co-culture, especially since lipid compositional remodeling is certainly possible and not unprecedented in symbioses (e.g., between plants and arbuscular mycorhizal fungi). We think it right to add a simple, honest sentence that states that insignificant changes in C18:3 vs.biomass are assumed for any physiological changes that might be experienced by M. elongata in coculture. (I think it probably is insignificant, but one should be precise and not speculate.) This applies to Figure 3—figure supplement 2 as well.

We agree, and we have added the discussion as suggested. Subsection “Nutrient -deficiency and benefits of co-cultivation for *N. oceanica* and *M. elongata*.”.

The nutrient exchange experiments are nicely performed and presented, but a discussion of their biological meaning is missing. M.elongata is a strong saprotroph and does not need the carbon coming from the alga. On the other hand, it also releases N to the alga. So, what is the benefit for the fungus? The only benefit seems to be the increased fungal biomass; however, this is contingent on the point noted above. The C18:3 quantification could mirror a different metabolism (lipid store) more than an active growth. Please add a point of discussion including the C and N conditions of the media in which these experiments were performed.

The co-culture of nutrient exchange experiments was performed in f/2 medium containing ^14^N (2.5 mM NaNO_3_) and ^12^C (20 mM NaHCO_3_ and ambient CO_2_). Thus, the alga should be the only resource of organic carbon for the fungus. Since *M. elongata* is a strong saprotroph, we did test whether the alga (as well as the fungus) was dead in the co-culture (Figure 2—figure supplement 3) and we also fed the fungus with heat-killed ^14^C-labeled alga and fed the alga with heat-killed ^14^C-labeled fungus. These results have been summarized in the Figure 2C and discussed in subsection “Carbon and Nitrogen Transfer between *N. oceanica* and *M. elongata*.”.

Subsection “Long-term co-cultivation leads to internalization of N. oceanica within M. elongata hyphae”: for algae that are inside live fungi, how can you be sure they are really alive based only on these data? Free/extracellular algae might be accessible to SYTOX Green, but algae in live fungi may never be exposed to SYTOX Green, yes? (The live fungi would exclude it, so one may not be able to tell intracellular viability of the algae.) We do not feel this is a major point because the EM data indicate that algae are alive, but it might not be possible to determine true viability of intracellular algae with exclusionary dyes and this should be acknowledged.

Excellent point and we acknowledge the short comings of this experiment. Our assumption that the algae inside the fungus are alive are mostly based on the fact that divide and to some extent that they are how chlorophyll fluorescence, which in the long run indicates an assembled photosynthetic membrane.

Abstract – Long term co-cultivation is a subjective term. The length of time observed should be inserted here.

The time has been added: “over a month”.

Figure 3—figure supplement 2 – Again, details of the statistical tests used should be added.

The details of the statistical tests have been added.

Figure 4 G -What is Mo? It is missing from the legend.

Mo stands for ‘*Mortierella* organelles’. This has been added in the legends of Figure 4G and Figure 4—figure supplement 2.

Subsection “Long-term co-cultivation leads to internalization of N. oceanica within M. elongata hyphae”: We suggest rewording to be more conservative in claims: "The results is consistent with the notion that both fungal host and algae inside are alive (Figure 4—figure supplement 5), although DIC microscopy…surrounded by fungal organelles and what appear to be lipid droplets…"Changed "is consistent with" for "showed", added "although" before "DIC microscopy, removed "the" before "fungal organelles", and changed "and what appear to be" for "such as". Lipid droplets are not fungal organelles…. As a side note: how does one know that these are lipid droplets? Do they stain with Nile Red? This is stated but not justified.

We have amended the text. We stained and visualized the lipid droplets with BODIPY by confocal microscopy (Figure 4—figure supplement 6). Lipid droplets are distinguishable because of their color (blue green) and size under light microscope.

Discussion section: We suggest rewording to be more conservative in claim: "green hyphae appear to remain alive after 2-months…". Changed "appear to remain" for "remained".

Amended as suggested.

Please insert the nature of the statistical tests used in the legends and/or in the Materials and methods section.

The statistical tests have been added in the legends as suggested.